# SE(3) Stochastic Flow Matching for Protein Backbone Generation

**Avishek (Joey) Bose**[1,2,3]*, **Tara Akhound-Sadegh**[1,2,3]*, **Guillaume Huguet**[4,2,3], **Kilian Fatras**[1,2,3], **Jarrid Rector-Brooks**[4,2,3], **Cheng-Hao Liu**[1,2,3], **Andrei Cristian Nica**[3], **Maksym Korablyov**[3], **Michael Bronstein**[5,3], **Alexander Tong**[4,2,3]

[1]McGill University, [2]Mila, [3]Dreamfold, [4]Université de Montréal, [5]University of Oxford

## Abstract

The computational design of novel protein structures has the potential to impact numerous scientific disciplines greatly. Toward this goal, we introduce FoldFlow a series of novel generative models of increasing modeling power based on the flow-matching paradigm over 3D rigid motions—i.e. the group SE(3)—enabling accurate modeling of protein backbones. We first introduce FoldFlow-Base a simulation-free approach to learning deterministic continuous-time dynamics and matching invariant target distributions on SE(3). We next accelerate training by incorporating Riemannian optimal transport to create FoldFlow-OT leading to the construction of both more simple and stable flows. Finally, we design FoldFlow-SFM coupling both Riemannian OT and simulation-free training to learn stochastic continuous-time dynamics over SE(3). Our family of FoldFlow generative models offers several key advantages over previous approaches to the generative modeling of proteins: they are more stable and faster to train than diffusion-based approaches, and our models enjoy the ability to map any invariant source distribution to any invariant target distribution over SE(3). Empirically, we validate FoldFlow on protein backbone generation of up to 300 amino acids leading to high-quality designable, diverse, and novel samples.

## 1 Introduction

Proteins are one of the basic building blocks of life. Their complex geometric structure enables specific inter-molecular interactions that allow for crucial functions within organisms, such as acting as catalysts in chemical reactions, transporters for molecules, and providing immune responses. Normally, such functions arise as a result of evolution. With the emergence of computational techniques, it has become possible to rationally design novel proteins with desired structures that program their functions. Such methods are now seen as the future of drug design and can lead to solutions to long-standing global health challenges. Some recent examples include rationally designed protein binders for receptors related to influenza (Strauch et al., 2017), COVID-19 (Cao et al., 2020a; Gainza et al., 2023), and cancer (Silva et al., 2019).

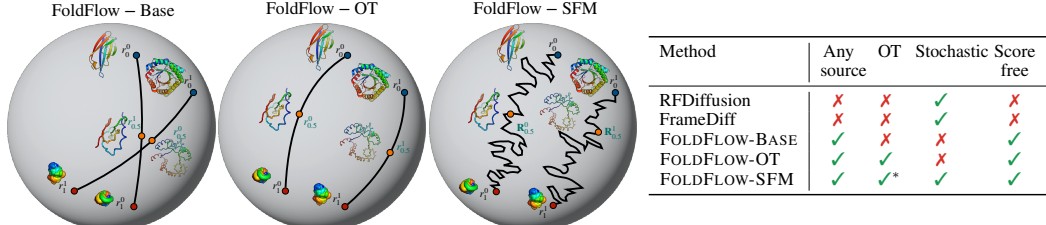

Figure 1: **Left:** Conditional probability paths learned by FoldFlow-Base (left), FoldFlow-OT (mid), and FoldFlow-SFM (right). We visualize the rotation trajectory of a single residue by the action of SO(3) on its homogenous space $\mathbb{S}^2$. **Right:** Table with the properties of each model: whether they can map from a general source distribution, perform optimal transport, are stochastic, or require the score of the $\mathcal{IG}_{\text{SO}(3)}$ density.

In protein engineering, the term *de novo* design refers to a setting when a new protein is designed to satisfy pre-specified structural and functional properties (Huang et al., 2016). Chemically, a protein is

---

*Equal Contribution. Corresponding authors: {`joey.bose,tara,alex`}`@dreamfold.ai`

a sequence of amino acids (*residues*) linked into a chain that folds into a complex 3D structure under the influence of electrostatic forces. The protein backbone can be seen as $N$ rigid bodies (corresponding to $N$ residues) that contain four heavy atoms $N - C_\alpha - C - O$. Mathematically, each residue can be associated with a frame that adheres to the symmetries of orientation-preserving rigid transformations (3D rotations and translations), forming the *special Euclidean group* $\mathrm{SE}(3)$ (Jumper et al., 2021); the entire protein backbone is described by the group product $\mathrm{SE}(3)^N$. The problem of protein design can be formulated as sampling from the distribution over this group, which is a perspective used in our paper. Recently, generative models have been generalized to Riemannian manifolds (Mathieu & Nickel, 2020; De Bortoli et al., 2022). However, they are not purpose-built to exploit the rich geometric structure of $\mathrm{SE}(3)^N$. Furthermore, several approaches require numerically expensive steps like simulating a Stochastic Differential Equation (SDE) during training or using the Riemannian divergence in the objective (Huang et al., 2022; Leach et al., 2022; Ben-Hamu et al., 2022).

**Our approach**. We introduce FOLDFLOW, a family of continuous normalizing flows (CNFs) tailored for distributions on $\mathrm{SE}(3)^N$ (fig. 1). We use the framework of Conditional Flow Matching (CFM), a *simulation-free* approach to learning CNFs by directly regressing time-dependent vector fields that generate probability paths (Lipman et al., 2022; Tong et al., 2023b). In particular, we introduce three new CFM-based models that learn $\mathrm{SE}(3)^N$-invariant distributions to generate protein backbones. In contrast to the previous $\mathrm{SE}(3)$ diffusion approach of Yim et al. (2023b), our FOLDFLOW is able to start from an informative prior. This enables new applications of generative models for protein design such as equilibrium conformation generation (Zheng et al., 2023).

**Main contributions**. Our first model FOLDFLOW-BASE extends the Riemannian flow matching approach (Chen & Lipman, 2023) by introducing a closed-form expression of the ground truth conditional vector field for $\mathrm{SO}(3)$ needed in the loss computation—thus greatly increasing speed and stability of training. Next, in FOLDFLOW-OT, we accelerate the training of our base model by constructing shorter and simpler flows using Riemannian Optimal Transport (OT) by proving the existence of a Monge map on $\mathrm{SE}(3)^N$. Finally, we present our most complex simulation-free model, FOLDFLOW-SFM, which learns a stochastic bridge on $\mathrm{SE}(3)^N$. Empirically, we validate our proposed models by learning to generate protein backbones of up to 300 residues. We observe that all FOLDFLOW models outperform the current SOTA non-pretrained diffusion model in FrameDiff (Yim et al., 2023b) for *in-silico* designability with FOLDFLOW-OT being the most designable. Moreover, for novelty FOLDFLOW-SFM is competitive with the current gold-standard RFDiffusion (Watson et al., 2023) with a fraction of the compute and data resources. We highlight the importance of novel and designable proteins as a key goal in AI-powered drug discovery where useful drug candidates are necessarily beyond the available training set (Schneider, 2018; Schneider et al., 2020; Marchand et al., 2022). Finally, we show the utility of FOLDFLOW on equilibrium conformation generation by learning to simulate molecular dynamics trajectories starting from a reference empirical distribution in comparison to an uninformed prior. Our code can be found at `https://github.com/DreamFold/FoldFlow`.

## 2 BACKGROUND AND PRELIMINARIES

### 2.1 RIEMANNIAN MANIFOLDS AND LIE GROUPS

**Riemannian manifolds**. Informally, an $n$-dimensional *manifold* $\mathcal{M}$ is a topological space locally equivalent (homeomorphic) to $\mathbb{R}^n$. This implies that one has the notion of 'neighbourhood' but not of 'distance' or 'angle' on $\mathcal{M}$. The manifold is said to be *smooth* if it additionally has a $C^\infty$ differential structure. At every point $x \in \mathcal{M}$, one can attach a *tangent space* $\mathcal{T}_x$. The disjoint union of tangent spaces forms the *tangent bundle*. A *Riemannian manifold*[1] $(\mathcal{M}, g)$ is additionally equipped with an inner product (*Riemannian metric*) $g_x : \mathcal{T}_x\mathcal{M} \times \mathcal{T}_x\mathcal{M} \to \mathbb{R}$ on the tangent space $\mathcal{T}_x\mathcal{M}$ at each $x \in \mathcal{M}$. The Riemannian metric $g$ allows to define key geometric quantities on $\mathcal{M}$ such as distances, volumes, angles, and length minimizing curves (*geodesics*). We consider functions defined on $\mathcal{M}$ and the tangent bundle, referred to as *scalar-* and *vector fields*, respectively. The *Riemannian gradient* is an operator $\nabla_g : C^\infty(\mathcal{M}) \to \mathfrak{X}(\mathcal{M})$ between the respective functional spaces. Given a smooth scalar field $f \in C^\infty(\mathcal{M})$, its gradient $\nabla_g f \in \mathfrak{X}(\mathcal{M})$ is the local direction of its steepest change.

**Lie groups**. A *Lie group* is a group that is also a differentiable manifold, in which the group operations $\circ : G \times G \to G$ of multiplication and inversion are smooth maps. It has a left action $L_h : G \to G$ defined by $x \mapsto h \circ x$ that is a topological isomorphism and whom the derivative is also an isomorphism between the tangent spaces on $G$. Since a group has an identity element, its tangent space is of special interest and is known as the *Lie algebra* $\mathfrak{G}$. The Lie algebra is a vector space with an associated bilinear

---

[1]We tacitly assume $\mathcal{M}$ to be orientable, connected, and complete and admit a volume form denoted as $dx$.

operation called the *Lie bracket* that is anticommutative and satisfies the Jacobi identity. Lie algebras elements can be mapped to the Lie group via the *exponential map* $\exp : \mathfrak{G} \to G$ which has an inverse called the logarithmic map $\log : G \to \mathfrak{G}$. For matrix Lie groups where the group action is the matrix multiplication, the $\exp$ and $\log$ maps correspond to the matrix exponential and matrix logarithm. The orientation-preserving rigid motions form the matrix Lie group $\mathrm{SE}(3) \cong \mathrm{SO}(3) \ltimes (\mathbb{R}^3, +)$, a semidirect product of rotations and translations (see §A.1 §A.2, and Hall (2013) for details).

## 2.2 Flow matching on Riemannian manifolds

Analogous to Euclidean spaces, probability densities can be defined on Riemannian manifolds as continuous non-negative functions $\rho : \mathcal{M} \to \mathbb{R}_+$ that integrate to $\int_{\mathcal{M}} \rho(x)dx = 1$.

**Probability paths on Riemannian manifolds**. Let $\mathbb{P}(\mathcal{M})$ be the space of probability distributions on $\mathcal{M}$. A *probability path* $\rho_t : [0, 1] \to \mathbb{P}(\mathcal{M})$ is an interpolation in probability space between two distributions $\rho_0, \rho_1 \in \mathbb{P}(\mathcal{M})$ indexed by a continuous parameter $t$. A one-parameter diffeomorphism $\psi_t : \mathcal{M} \to \mathcal{M}$ is known as a *flow* on $\mathcal{M}$ and is defined as the solution of the following ordinary differential equation (ODE): $\frac{d}{dt}\psi_t(x) = u_t(\psi_t(x))$, with initial conditions $\psi_0(x) = x$, for $u : [0, 1] \times \mathcal{M} \to \mathcal{M}$ a time-dependent smooth vector field. We say the flow $\psi_t$ generates $\rho_t$ if it pushes forward $\rho_0$ to $\rho_1$ by following the time-dependent vector field $u_t$—i.e. $\rho_t = [\psi_t]_\#(\rho_0)$. As $\psi_t$ is a diffeomorphism, $\rho_t$ verifies the famous *continuity equation* and the density can be calculated using the instantaneous change of variables formula for Riemannian manifolds (Mathieu & Nickel, 2020).

**Riemannian flow matching**. Given a probability path $\rho_t$ that connects $\rho_0$ to $\rho_1$, and its associated flow $\psi_t$, we can learn a CNF by directly regressing the vector field $u_t$ with a parametric one $v_\theta \in \mathfrak{X}(\mathcal{M})$. This technique is termed *flow matching* (Lipman et al., 2022, FM) and leads to a simulation-free training objective as long as $\rho_t$ satisfies the boundary conditions $\rho_0 = \rho_{\mathrm{data}}$ and $\rho_1 = \rho_{\mathrm{prior}}$. Unfortunately, the vanilla flow matching objective is intractable as we generally do not have access to the closed-form of $u_t$ that generates $\rho_t$. Instead, we can opt to regress $v_\theta$ against a conditional vector field $u_t(x_t|z)$, generating a conditional probability path $\rho_t(x_t|z)$, and use it to recover the target unconditional path: $\rho_t(x_t) = \int_{\mathcal{M}} \rho_t(x_t|z)q(z)dz$. The vector field $u_t$ can also be recovered by marginalizing of conditional vector fields: $u_t(x) := \int_{\mathcal{M}} u_t(x|z)\frac{\rho_t(x_t|z)q(z)}{\rho_t(x)}dz$. The Riemannian CFM objective (Chen & Lipman, 2023) is then,

$$\mathcal{L}_{\mathrm{rcfm}}(\theta) = \mathbb{E}_{t,q(z),\rho_t(x_t|z)}\|v_\theta(t, x_t) - u_t(x_t|z)\|_g^2, \quad t \sim \mathcal{U}(0, 1). \tag{1}$$

As FM and CFM objectives have the same gradients (Tong et al., 2023b), at inference, we can generate by sampling from $\rho_1$, and using $v_\theta$ to propagate the ODE backward in time.

## 2.3 Protein Backbone Parametrization

Our protein backbone parameterization follows the seminal work of AlphaFold2 (Jumper et al., 2021, AF2) in that we associate a frame with each residue in the amino acid sequence. For a protein of length $N$ this results in $N$ frames that are $\mathrm{SE}(3)$-equivariant. Each frame maps a rigid transformation starting from idealized coordinates of four heavy atoms $\mathrm{N}^*, \mathrm{C}_\alpha^*, \mathrm{C}^*, \mathrm{O}^* \in \mathbb{R}^3$, with $\mathrm{C}_\alpha^* = (0, 0, 0)$ being

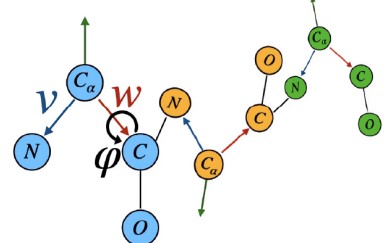

Figure 2: Protein backbone parametrization.

centered at the origin, and is a measurement of experimental bond angles and lengths (Engh & Huber, 2012). Thus, residue $i \in [N]$ is represented as an action of $x^i = (r^i, s^i) \in \mathrm{SE}(3)$ applied to the idealized frame $[\mathrm{N}, \mathrm{C}_\alpha, \mathrm{C}, \mathrm{O}]^i = x^i \circ [\mathrm{N}^*, \mathrm{C}_\alpha^*, \mathrm{C}^*, \mathrm{O}^*]$. To construct the backbone oxygen atom O, we rotate about the axis given by the bond between $\mathrm{C}_\alpha$ and C using an additional rotation angle $\varphi$. Finally, we denote the full 3D coordinates of all heavy atoms as $\mathrm{A} \in \mathbb{R}^{N \times 4 \times 3}$. An illustration for this backbone parametrization is provided in fig. 2 with rotations being parametrized as $r = v \times w$.

## 3 FoldFlow for condition flow matching on $\mathrm{SE}(3)$

We seek to learn an $\mathrm{SE}(3)^N$ invariant density $\rho_t$ by training a flow using the objective in eq. (1). To do so we can pushforward an $\mathrm{SE}(3)^N$-invariant source distribution $\rho_1$ to the empirical distribution of proteins $\rho_0$ using an equivariant flow. One way to guarantee the existence of a translation-invariant measure is to construct a subspace that is invariant to global translations. This can be achieved by simply subtracting the center of mass of all inputs to the flow (Rudolph et al., 2021; Yim et al., 2023b). Formally, this leads to an invariant measure on $\mathrm{SE}(3)_0^N$ which is a subgroup of $\mathrm{SE}(3)^N$. We then note that $\mathrm{SE}(3)_0^N$ is a product group and thus the Riemannian metric extends in a natural way to the product

space: the exponential and logarithmic maps decompose across each manifold, and the geodesic distance in $\mathrm{SE}(3)_0^N$ is simply the sum of each individual distance in the product. As such, a flow on $\mathrm{SE}(3)_0^N$ can be built from separate flows for each residue in the backbone, on $\mathrm{SE}(3)$, after centering.

**Decomposing** $\mathrm{SE}(3)$ **into** $\mathrm{SO}(3)$ **and** $\mathbb{R}^3$. As Lie groups are manifolds, they can be equipped with a metric to obtain a Riemannian structure. In the case of $\mathrm{SE}(3) \cong \mathrm{SO}(3) \ltimes (\mathbb{R}^3, +)$ there are multiple possible choices, but a natural one is $\langle \mathfrak{x}, \mathfrak{x}' \rangle_{\mathrm{SE}(3)} = \langle \mathfrak{r}, \mathfrak{r}' \rangle_{\mathrm{SO}(3)} + \langle s, s' \rangle_{\mathbb{R}^3}$ (see §A.2). Moreover, the disintegration of measures implies that every $\mathrm{SE}(3)$-invariant measure can be broken down to a $\mathrm{SO}(3)$-invariant measure and a measure proportional to the Lebesgue measure on $\mathbb{R}^3$ (Pollard, 2002). Thus, we may simply build independent flows on $\mathrm{SO}(3)$ and $\mathbb{R}^3$. In this section, we focus on designing FOLDFLOW models on $\mathrm{SO}(3)$, as CFMs on $\mathbb{R}^d$ are well-studied in Albergo & Vanden-Eijnden (2023); Lipman et al. (2022); Tong et al. (2023b) (we provide a complete description in §B). We use the notation $\rho_t$, $q$, and $\pi$ for densities whose support is determined by its context.

### 3.1 FOLDFLOW-BASE

To construct a flow on $\mathrm{SO}(3)$ that connects the target distribution $\rho_0$ to a source distribution $\rho_1$, we must first choose a parametrization of the group elements. The most familiar and natural parametrization is by orthogonal matrices with unit determinant (see §A.1 for a discussion on other parametrizations e.g. rotation-vector). The Lie algebra $\mathfrak{so}(3)$ contains skew-symmetric matrices $\mathfrak{r}$ that are tangent vectors at the identity of $\mathrm{SO}(3)$. The last important component that we require is a choice of Riemannian metric for $\mathrm{SO}(3)$. A canonical bi-invariant metric for $\mathrm{SO}(3)$ can be derived from the Killing form (see §A.1), and is given by: $\langle \mathfrak{r}, \mathfrak{r}' \rangle_{\mathrm{SO}(3)} = \mathrm{tr}(\mathfrak{r}\mathfrak{r}'^T)/2$.

$\mathrm{SO}(3)$ **conditional vector fields and flows**. We seek to construct a conditional vector field $u_t(r_t | z)$, lying on the tangent space $\mathcal{T}_{r_t}\mathrm{SO}(3)$, that transports $r_0 \sim \rho_0$ to $r_1 \sim \rho_1$. Following, Tong et al. (2023b) we set the conditioner to $z = (r_0, r_1)$. Next, we construct a flow $\psi_t$ that connects $r_0$ to $r_1$. We follow the most natural strategy which is to build the flow using the geodesic between $r_0$ and $r_1$. For general $\mathcal{M}$, the geodesic interpolant between two points, indexed by $t$, has the following form:

$$r_t = \exp_{r_0}(t \log_{r_0}(r_1)). \tag{2}$$

For rotation matrices, eq. (2) involves computing the $\exp$ and $\log$ maps which are both infinite matrix power series. Unfortunately, controlling the approximation error of $\log_{r_0}$ map is computationally expensive as the de facto inverse scaling method for computing matrix logarithms requires estimating and calculating fractional matrix powers (Al-Mohy & Higham, 2012). Instead, we use a numerical trick by converting $r_1$ to its axis-angle representation which gives a vector representation of $\mathfrak{r}_1 \in \mathfrak{so}(3)$ and, by definition, lives at the tangent space at the identity and is equivalent to $\log_e(r_1)$. Next, we can parallel transport $\mathfrak{r}_1$ to the tangent space of $r_0$ since Lie algebras of all tangent spaces are isomorphic and $\mathrm{SO}(3)$ carries a free action which gives us the desired end result $\log_{r_0}(r_1)$.

Given $r_t$, we can build constant velocity vector fields by directly leveraging the ODE associated with the conditional flow: $\frac{d}{dt}\psi_t(r) = \dot{r}_t$ (Chen & Lipman, 2023). As a result, computing $u_t(r_t | z)$ boils down to computing the point $r_t$ along the ODE and taking its time derivative. In practice, taking the time-derivative to compute $u_t = \dot{r}_t$ amounts to using autograd to compute the gradient during a forward pass. We can overcome this unnecessary overhead without relying on automatic differentiation but instead by exploiting the geometry of the problem. Specifically, we calculate the $\mathfrak{so}(3)$ element corresponding to the relative rotation between $r_0$ and $r_t$, given by $r_t^\top r_0$. We divide by $t$ to get a vector which is an element of $\mathfrak{so}(3)$ and corresponds to the skew-symmetric matrix representation of the velocity vector pointing towards the target $r_1$. Finally, we parallel-transport the velocity vector to the tangent space $\mathcal{T}_{r_t}\mathrm{SO}(3)$ using left matrix multiplication by $r_t$. These operations can be concisely written as $\log_{r_t}(r_0)$, where we use our numerical trick to calculate the matrix logarithm. The closed form expression of the loss to train the $\mathrm{SO}(3)$ component of FOLDFLOW-BASE is thus

$$\mathcal{L}_{\text{FOLDFLOW-BASE}-\mathrm{SO}(3)}(\theta) = \mathbb{E}_{t \sim \mathcal{U}(0,1), q(r_0, r_1), \rho_t(r_t | r_0, r_1)} \left\| v_\theta(t, r_t) - \log_{r_t}(r_0)/t \right\|_{\mathrm{SO}(3)}^2. \tag{3}$$

In eq. (3) the conditioning distribution $q(z) = q(r_0, r_1)$ is the independent coupling $q(r_0, r_1) = \rho_0 \rho_1$, where $\rho_1 = \mathcal{U}(\mathrm{SO}(3))$ and is left-invariant w.r.t. to the Haar measure. Also, note that the vector field in eq. (3) is on the tangent space $v_\theta \in \mathcal{T}_{r_t}\mathrm{SO}(3)$ and the norm is induced by the metric on $\mathrm{SO}(3)$.

### 3.2 FOLDFLOW-OT

The conditional vector field $u_t(r_t | z)$ generates the conditional probability path $\rho_t(r_t | z)$ which deterministically evolves $\rho_0$ to $\rho_1$. However, there is no reason to believe the conditional probability

path is *optimal* in the sense that it is a length-minimizing curve, under an appropriate metric, in the space of distributions $\mathbb{P}(\mathrm{SO}(3))$. We seek to rectify this by constructing conditional probability paths that are not only shorter and straighter, but also more stable from an optimization perspective. This is motivated by previous research (Tong et al., 2023b; Pooladian et al., 2023a) which has shown optimal transport to lead to faster training with a lower variance training objective in Euclidean spaces.

To this end, we propose FOLDFLOW-OT, a model that accelerates FOLDFLOW-BASE by constructing conditional probability paths using *Riemannian optimal transport*. The interpolation measure $\rho_t$ connects $\rho_0 \to \rho_1$ and is built from Riemannian OT which solves the Monge optimal transport problem:

$$\mathrm{OT}(\rho_0, \rho_1) = \inf_{\Psi : \Psi_{\#}\rho_0 = \rho_1} \int_{\mathrm{SE}(3)_0^N} \frac{1}{2} c(x, \Psi(x))^2 \, d\rho_0(x). \tag{4}$$

Here $c$ is the geodesic cost induced by the metric (cf. eq. (25) in §A.2) and $\Psi$ a pushforward map: $\rho_0 \to \rho_1$. A related problem, called the OT-Kantorovich formulation, relaxes the Monge problem by looking for a joint probability distribution $\pi$ minimizing the displacement cost of transporting $\rho_0$ to $\rho_1$ (see §C.1). The uniqueness of the Monge map over $\mathrm{SE}(3)_0^N$ is guaranteed under some assumptions on the measures $\rho_0, \rho_1$ as stated in the following proposition and proven in §C.2.

> **Proposition 1.** *Let us consider $\mathrm{SE}(3)_0^N$ with the product distance $d_{\mathrm{SE}(3)_0^N}$ and two compactly supported probability distributions $\rho_0, \rho_1 \in \mathbb{P}(\mathrm{SE}(3)_0^N)$. In addition, suppose that $\rho_0$ is absolutely continuous with respect to Riemannian volume form (i.e., $\rho_0 \ll dx$). Then for the distance $c = \frac{1}{2} d_{\mathrm{SE}(3)_0^N}^2$, the Kantorovich and Monge problems admit a unique solution that is connected as follows $\pi = (id \times \Psi)_{\#}\rho_0$, where $\Psi$ is almost uniquely determined everywhere $\rho_0$. Furthermore, we have that $\Psi(x) = \exp_x(\nabla\phi(x))$ for some $d_{\mathrm{SE}(3)_0^N}^2$-concave function $\phi$.*

Following this proposition, we define the McCann interpolants as $\rho_t(x) = (\exp_x(-t\nabla\phi(x)))_{\#}\rho_0$. While it is possible to approximate the Monge map and McCann interpolants using $c$-concave functions, it imposes practical limitations on the architecture of the flow (Cohen et al., 2021). Instead, we use the correspondence between the Monge and Kantorovich problems and rely on the optimal transport plan $\pi$. Formally, we draw two samples from $q(z) = q(x_0, x_1) := \pi(x_0, x_1)$ and we compute for a given frame $\rho_t(r_t|r_0, r_1) = \delta(\exp_{r_0}(t\log_{r_0}(r_1)))$, where $\delta$ is a Dirac. Since our choice of metric for $\mathrm{SE}(3)$ factorizes into metrics on $\mathrm{SO}(3)$ and $\mathbb{R}^3$, we can use independent losses on rotations and translations—similar to FOLDFLOW-BASE—and repeat this over $N$ frames, as long as each geometric quantity in $\pi$ is coupled properly. Defining $\bar{\pi}(r_0, r_1)$ as the projection of $\pi(x_0, x_1)$ on $\mathrm{SO}(3)$, we present the $\mathrm{SO}(3)$ loss for a single frame in FOLDFLOW-OT as

$$\mathcal{L}_{\mathrm{FOLDFLOW-OT-SO(3)}}(\theta) = \mathbb{E}_{t \sim \mathcal{U}(0,1), \bar{\pi}(r_0, r_1), \rho_t(r_t|r_0, r_1)} \left\| v_\theta(t, r_t) - \log_{r_t}(r_0)/t \right\|_{\mathrm{SO}(3)}^2. \tag{5}$$

## 3.3 FOLDFLOW-SFM

We finally present FOLDFLOW-SFM, which builds on the foundations of both FOLDFLOW-BASE and FOLDFLOW-OT. Departing from the deterministic dynamics of the previous models, we aim to build a *stochastic* flow over $\mathrm{SE}(3)_0^N$ by replacing these deterministic bridges with guided stochastic bridges. Previous research has shown that, compared to ODEs, SDEs have the important benefit of being more robust to noise in high dimensions (Tong et al., 2023a; Shi et al., 2023; Liu et al., 2023a). For proteins in $\mathrm{SE}(3)_0^N$, this means the generative process has greater empirical calibre to sample *outside* the support of the training distribution—crucial for generating designable proteins that are also *novel*. In Euclidean space, we can build a translation invariant flow on $(\mathbb{R}^3)_0^N$ by using a (reverse time) Brownian bridge as the conditional flow between the points,

$$d\mathrm{S}_t = \frac{\mathrm{S}_t - s_0}{t} dt + \gamma(t)d\mathrm{W}_t, \quad \mathrm{S}_1 = s_1. \tag{6}$$

This flow, also known as Doob's h-transform (Doob, 1984), is easy to sample from in a simulation-free manner and correctly maps between arbitrary samplable marginals in expectation. Specifically, we can build a simulation-free bridge by sampling from the conditional probability $\rho_t(x_t|x_0, x_1) = \mathcal{N}(x_t; tx_1 + (1-t)x_0, \gamma^2 t(1-t))$ (Shi et al., 2023; Albergo et al., 2023). See §B for more details.

**Brownian Bridge on** $\mathrm{SO}(3)$. On $\mathrm{SO}(3)$, we aim to model the dynamics between rotations matrices by a guided diffusion bridge (Jensen et al., 2022; Liu et al., 2022), leading to the following dynamics,

$$d\mathrm{R}_t = \frac{\log_{\mathrm{R}_t} r_0}{t} dt + \gamma(t)d\mathrm{B}_t, \quad \mathrm{R}_1 = r_1, \tag{7}$$

for $B_t$ the Brownian motion on $\mathrm{SO}(3)$ (see §D.1 for further technical details on this SDE). Despite the close resemblance of this SDE to the translation one in eq. (6), the corresponding Brownian bridge—to the best of our knowledge—does not have a closed-form expression for $\rho_t(r_t|r_0, r_1)$. Thus, to sample from $\rho_t(r_t|r_0, r_1)$ correctly, we start at $r_1$ and simulate the bridge backward in time using algorithm 3 in §I.2. Given this form of the conditional bridge, we can make use of a flow-matching loss to optimize a flow between source and target distributions on $\mathrm{SO}(3)$ as follows:

$$\mathcal{L}_{\mathrm{SFM-SO(3)}}(\theta) = \mathbb{E}_{t \sim \mathcal{U}(0,1), \bar{\pi}(r_0, r_1), \rho_t(\tilde{r}_t|r_0, r_1)} \left\| v_\theta(t, \tilde{r}_t) - \log_{\tilde{r}_t}(r_0)/t \right\|_{\mathrm{SO(3)}}^2. \tag{8}$$

Here $\tilde{r}_t$ is a sample from the bridge between $r_0$ and $r_1$. When $\pi(x_0, x_1)$ is a valid coupling between $\rho_0$ and $\rho_1$—and thus $\bar{\pi}(r_0, r_1)$ on $\mathrm{SO}(3)$—this objective is equivalent in expectation to matching directly the (computationally intractable) marginal loss $\mathcal{L}_{\mathrm{USFM}} = \mathbb{E}_{t \sim \mathcal{U}(0,1), \rho_t(\tilde{r})} \left\| v_\theta(t, \tilde{r}_t) - u(t, \tilde{r}_t) \right\|_{\mathrm{SO(3)}}^2$. The correctness of this approach is established in the next proposition and proved in §D.3.

**Proposition 2.** *Given $\rho_t(x) > 0$, $\forall x \in \mathrm{SE}(3)_0^N$, the conditional and unconditional* FOLDFLOW-SFM *losses have equal gradients* w.r.t. $\theta$: $\nabla_\theta \mathcal{L}_{\mathrm{USFM}}(\theta) = \nabla_\theta \mathcal{L}_{\mathrm{SFM}}(\theta)$.

This result allows us to learn a stochastic flow from any source to any target distribution supported on $\mathrm{SE}(3)_0^N$, only requiring samples from both distributions. However, it does require simulation of an SDE to sample from the conditional probability $\rho_t(r_t|r_0, r_1)$, limiting scalability.

**An Efficient Simulation-free Approximation**. Unfortunately, sampling from the correct conditional bridge requires simulation and is thus computationally expensive for training. In practice, we use a simulation-free approximation that closely matches the true conditional probability path on $\mathrm{SO}(3)$, $\rho_t(\tilde{r}_t|r_0, r_1)$. Specifically, we approximate $\rho_t$ with the simulation-free alternative,

$$\hat{\rho}_t(\tilde{r}_t|r_0, r_1) = \mathcal{IG}_{\mathrm{SO(3)}}\left(\tilde{r}_t; \exp_{r_0}(t \log_{r_0}(r_1)), \gamma^2(t)t(1-t)\right), \tag{9}$$

where $\mathcal{IG}_{\mathrm{SO(3)}}$ denotes the isotropic Gaussian distribution on $\mathrm{SO}(3)$. This distribution can be seen as an analog of the Gaussian distribution in $\mathbb{R}^d$. It is the heat kernel on $\mathrm{SO}(3)$ (Nikolayev & Savyolov, 1900) and it can be seen as the limit of small i.i.d. rotations in 3D (Qiu, 2013). Additionally, it has some of the desirable properties of the normal distribution, such as being closed under convolution (Nikolayev & Savyolov, 1900). We provide both the training and sampling algorithms for FOLDFLOW-SFM in §I, details on $\mathcal{IG}_{\mathrm{SO(3)}}$ in §A.3, and results on the approximation error in §D.2.

## 4 MODELING PROTEIN BACKBONES USING FOLDFLOW

To model protein backbones using FOLDFLOW models we parameterize the velocity prediction $v_\theta(t, x_t)$ as a function that consumes a protein $x_t$ on the conditional path at time $t$ and predicts the starting point $\hat{x}_0$. Specifically, the predicted velocity is $v_\theta(t, x_t) = \nabla_g d(\hat{x}_0, x_t)^2/t$, with $\hat{x}_0 = w_\theta(t, x_t)$. This choice of parameterization has two principal benefits. (1) It allows the usage of specialized architectures specifically designed for structure prediction, and (2) it allows for auxiliary protein-specific losses to be placed directly on the $\hat{x}_0$ to improve performance.

**Architecture**. Following by Anand & Achim (2022); Yim et al. (2023b) (FrameDiff) we use the structure module of AF2 to model $w_\theta$. This begins with a time-dependent node and edge embeddings $N_\theta(t, x_t)$ and $E_\theta(t, x_t)$, followed by layers of invariant point attention. We use a small MLP head on top of the node embeddings to predict the torsion angle of the oxygen $\varphi$ as $\hat{\varphi} = \mathrm{MLP}(N_\theta(t, x_t))$.

**Full Loss**. Our FOLDFLOW models are trained to optimize a flow-matching loss on $\mathrm{SO}(3)$ and $\mathbb{R}^3$ for each residue $i \in [N]$ in the backbone. These are denoted $\mathcal{L}_{\mathrm{FOLDFLOW-SO(3)}}$ and $\mathcal{L}_{\mathrm{FOLDFLOW-}\mathbb{R}^3}$ (see §B for the complete expression in $\mathbb{R}^3$) respectively. In addition to the flow-matching losses, we also include auxiliary losses from Yim et al. (2023b) which enforce good predictions at the atomic level in the $\mathbb{R}^3$ atomic representation $A$. These include a direct regression on the backbone (bb) positions $\mathcal{L}_{\mathrm{bb}}$ and a loss on the pairwise atomic distance in a local neighbourhood $\mathcal{L}_{\mathrm{2D}}$,

$$\mathcal{L}_{\mathrm{aux}} = \mathbb{E}_{\mathcal{Q}}\left[\mathcal{L}_{\mathrm{bb}} + \mathcal{L}_{\mathrm{2D}}\right], \quad \mathcal{L}_{\mathrm{bb}} = \frac{1}{4N}\sum \|A_0 - \hat{A}_0\|^2, \quad \mathcal{L}_{\mathrm{2D}} = \frac{\|\mathbf{1}\{D < 6\text{Å}\}(D - \hat{D})\|^2}{\sum \mathbf{1}_{D < 6\text{Å}} - N}$$

Here $\mathcal{Q}(t, x_0, x_1, \tilde{x}_t) := \mathcal{U}(0, 1) \otimes \bar{\pi}(x_0, x_1) \otimes \rho_t(\tilde{x}_t|x_0, x_1)$ is the factorized joint distribution, $\mathbf{1}$ is the indicator function, $A$ is in Angstroms (Å), $D$ is an $N \times N \times 4 \times 4$ tensor containing the pairwise distances between the four heavy atoms, i.e. $D_{ijab} = \|A_{ia} - A_{jb}\|$, and $\hat{D}$ is defined similarly from $\hat{A}$. We only apply auxiliary losses for $t < 0.25$, with scaling $\lambda_{\mathrm{aux}}$ for a final loss of:

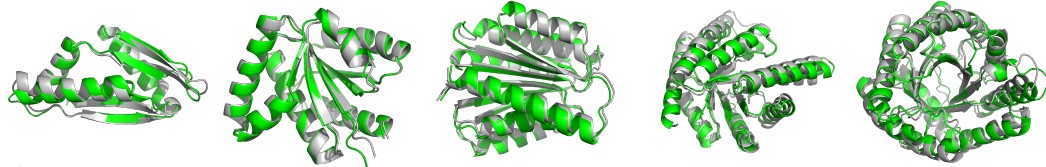

Figure 3: FOLDFLOW-SFM generated structures in green compared to ProteinMPNN $\longrightarrow$ ESMFold refolded structures in grey. Samples with RMSD < 2Å for lengths 100, 150, 200, 250, 300 from left to right.

$$\mathcal{L}_{\text{FOLDFLOW}}(\theta) = \mathcal{L}_{\text{FOLDFLOW}-SO(3)} + \mathcal{L}_{\text{FOLDFLOW}-\mathbb{R}^3} + \mathbf{1}\{t < 0.25\}\lambda_{\text{aux}}\mathcal{L}_{\text{aux}}. \tag{10}$$

## 5 EXPERIMENTS

### 5.1 SO(3) SYNTHETIC DATA

We evaluate all our FOLDFLOW models on synthetic multimodal densities on SO(3) as done by Brofos et al. (2021) (see §F.1 for details). We report the Wasserstein distance between generated and ground truth samples in table 1 and visualize the generated samples in fig. 7 in §F.2. We find that all our proposed methods correctly model all the modes of the ground truth distribution. However, FOLDFLOW-BASE exhibits mode shrinkage in relation to the ground truth. FOLDFLOW-OT, FOLDFLOW-SFM,

Table 1: Mean and std of the 1- and 2-Wasserstein distances, computed against 5000 points in the test set, over 5 seeds.

| $(\mu \pm \sigma)$ | $\mathcal{W}_1 (\times 10^{-2})$ | $\mathcal{W}_2 (\times 10^{-1})$ |
|---|---|---|
| FOLDFLOW-BASE | $5.39 \pm 0.88$ | $1.52 \pm 0.27$ |
| FOLDFLOW-OT | $4.96 \pm 0.27$ | $1.25 \pm 0.12$ |
| FOLDFLOW-SFM | $4.92 \pm 1.56$ | $1.26 \pm 0.49$ |
| Simulated SDE | $5.13 \pm 1.36$ | $1.33 \pm 0.44$ |

and the simulated SDE results in comparable performance, with the OT-based method being the best. Importantly, this shows that our simulation-free approximation of the SDE does not hinder model performance, and combined with its significant speedup justifies its use in protein experiments.

### 5.2 PROTEIN BACKBONE DESIGN

Table 2: Comparison of Designability (fraction of proteins with scRMSD < 2.0Å and mean scRMSD), Diversity (avg. pairwise TMscore), Novelty (max. TM-score to PDB and fraction of proteins with averaged max. TMscore < 0.5 and scRMSD < 2.0Å). Designability and Novelty metrics include standard errors.*RFDiffusion and Genie have larger training sets that likely overestimate novelty with respect to our dataset.

| | Designability | | Novelty | | Diversity ($\downarrow$) | iters / sec ($\uparrow$) |
|---|---|---|---|---|---|---|
| | Fraction ($\uparrow$) | scRMSD ($\downarrow$) | Fraction ($\uparrow$) | avg. max TM ($\downarrow$) | | |
| RFDiffusion | $0.969 \pm 0.023$ | $0.650 \pm 0.136$ | *$0.708 \pm 0.060$ | *$0.449 \pm 0.012$ | 0.256 | — |
| Genie | $0.581 \pm 0.064$ | $2.968 \pm 0.344$ | *$0.556 \pm 0.093$ | *$0.434 \pm 0.016$ | 0.228 | — |
| FrameDiff-ICML | $0.402 \pm 0.062$ | $3.885 \pm 0.415$ | $0.176 \pm 0.124$ | $0.542 \pm 0.046$ | 0.237 | — |
| FrameDiff-Improved | $0.555 \pm 0.071$ | $2.929 \pm 0.354$ | $0.296 \pm 0.112$ | $0.457 \pm 0.026$ | 0.278 | — |
| FrameDiff-Retrained | $0.612 \pm 0.060$ | $2.990 \pm 0.307$ | $0.108 \pm 0.083$ | $0.684 \pm 0.032$ | 0.403 | 1.278 |
| FOLDFLOW-BASE | $0.657 \pm 0.042$ | $3.000 \pm 0.271$ | $0.432 \pm 0.074$ | $0.452 \pm 0.024$ | 0.264 | 2.674 |
| FOLDFLOW-OT | $0.820 \pm 0.037$ | $1.806 \pm 0.249$ | $0.484 \pm 0.068$ | $0.460 \pm 0.020$ | 0.247 | 2.673 |
| FOLDFLOW-SFM | $0.716 \pm 0.040$ | $2.296 \pm 0.391$ | $0.544 \pm 0.061$ | $0.411 \pm 0.023$ | 0.248 | 2.647 |

We evaluate FOLDFLOW models in generating valid, diverse, and novel backbones by training on a subset of the Protein Data Bank (PDB) with 22,248 proteins. We compare FOLDFLOW to pretrained versions of FrameDiff (Yim et al., 2023b) (FrameDiff-ICML), the improved version on the authors' GitHub (FrameDiff-Improved), Genie (Lin & AlQuraishi, 2023), and RFDiffusion, which is the gold standard (Watson et al., 2023). We also retrain FrameDiff (FrameDiff-Retrained) on our dataset, which contains ∼10% more admissible structures, while inheriting the majority of the hyperparameters of FOLDFLOW. We provide a detailed description of all the metrics in §I.6. Figures 3 and 10 visualize generated samples and ESM-refolded structures.

We report our findings in table 2 and observe that FOLDFLOW outperforms FrameDiff-Retrained on all three metrics. We identify FrameDiff as the most comparable baseline as it is the current SOTA model that does not utilize pre-training while using comparable resources. In contrast to FOLDFLOW we highlight that RFDiffusion uses a pre-trained backbone and a significantly larger model ($60m$ vs. $17m$ parameters), training set, and compute resources (1800 vs. 10 GPU days). We also note that Genie is trained on a larger dataset ($195k$ vs. $22k$), which hinders rigorous comparisons with FOLDFLOW. Next, we analyze the performance of FOLDFLOW on each metric in detail.

**Designability**. We measure designability using the *self-consistency* metric with ProteinMPNN (Dauparas et al., 2022) and ESMFold (Lin et al., 2022), counting the fraction of proteins that refold ($C_\alpha$-RMSD (scRMSD) < 2.0Å and mean scRMSD) over 50 proteins at lengths $\{100, 150, 200, 250, 300\}$.

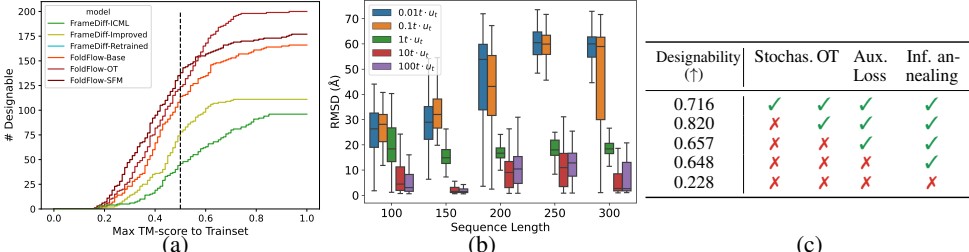

Figure 4: (a) Count of novel and designable (scRMSD < 2) proteins out of 250 at various novelty thresholds with TM-score to the training set < $x$ for models trained on subsets of PDB. (b) scRMSD of designed proteins vs. ESMFold under flow scaling (c) Table showing an ablation study of FOLDFLOW features against designability.

In table 2, we find that all FOLDFLOW models achieve significantly higher Frac. designability score than all FrameDiff models, and appreciably close the gap to RFDiffusion, e.g. $\Delta = 0.149$ vs. $\Delta = 0.357$ for FOLDFLOW-OT and FrameDiff-Retrained respectively. When retrained on our dataset with 10% more samples, we find that FrameDiff is more designable, but is still below all FOLDFLOW models. We also note that while FOLDFLOW-OT creates the highest fraction of designable proteins (excluding RFDiffusion), it has relatively low diversity and novelty. We find that adding stochasticity with FOLDFLOW-SFM results in a model that beats FrameDiff-Improved on every metric and can dramatically improve novelty at the cost of worse designability (table 2). In fig. 9 in §G.1, we plot designability versus sequence length and observe the largest gains on sequence lengths < 300.

**Diversity**. We use the average pairwise TM-score of the *designable* generated samples averaged across lengths as our diversity metric (lower is better). We find an inverse correlation between performance on designability and diversity metrics for FrameDiff models, which interestingly does not hold for FOLDFLOW models. We note that FOLDFLOW models have comparable diversity to the baselines with FOLDFLOW-OT and FOLDFLOW-SFM being the most diverse.

**Novelty**. Designing novel but realistic protein structures compared to the training data is also an important goal. Unlike conventional generative modeling problems, e.g. images, the novelty of proteins is particularly important since the entire premise of ML-driven drug discovery requires developing *original* drugs that may be vastly different than current human knowledge (training data) but also synthesizable (designable) (Marchand et al., 2022; Schneider et al., 2020; Schneider, 2018). We measure novelty using two metrics: 1.) the fraction of designable proteins with TM-score < 0.5 as used in Lin & AlQuraishi (2023) (higher is better) and 2.) the average maximum TM-score of designable generated proteins to the training data (lower is better). In fig. 4a count the number of designable proteins as a function of the Max TM-score to the training set. We see that FOLDFLOW-SFM designs the most novel structures against all methods including RFDiffusion and Genie. This substantiates the hypothesis that the stochasticity of the learned SDE is crucial to the improved robustness in high dimensions of FOLDFLOW-SFM versus FOLDFLOW-BASE, and FOLDFLOW-OT which allows it to sample designable proteins far outside of the support of the training set.

**Inference Annealing**. We now describe a numerical trick during inference that greatly improves the designability of FOLDFLOW which we term *inference annealing*. Instead of following the theoretical ODE or SDE for generating rotations, we use a multiplicative scaling of the velocity—e.g. $dR_t = i(t)v_\theta(t, R_t) dt + \gamma(t)dB_t$ for some positive function $i(t)$. In practice, we use $i(t) = ct$ for some constant $c$. This annealing removes an unwanted increase in the flow norm during the end of inference (fig. 8). We observe that larger $c$ drastically increases the designability of FOLDFLOW (fig. 4b). In practice, we use values of $c \approx 10$, which leads to designable yet diverse structures.

**Ablation study of FOLDFLOW**. Next, we ablate various additions to the FOLDFLOW-BASE model in fig. 4c and report the full extended ablation in table 6. We find FOLDFLOW-OT creates the most designable model, but adding stochasticity helps increase novelty and diversity. We also find that inference annealing is critical to the performance of the model in terms of achieving higher designability.

## 5.3 EQUILIBRIUM CONFORMATION GENERATION

Table 3: $\mathcal{W}_2$ in angle space between generated and test samples.

| | $\mathcal{W}_2$ | $\mathcal{W}_2$@56 |
|---|---|---|
| FOLDFLOW | 4.379 | 0.406 |
| FOLDFLOW-Rand | 4.446 | 0.557 |
| FrameDiff | 4.844 | 0.800 |

Modeling various protein conformations is crucial in determining biological behaviours such as mechanisms of actions or binding affinity to other proteins. Unlike diffusion models, FOLDFLOW can easily be instantiated from any sampleable source distribution. To test this, we model the equilibrium distribution of a protein given initial predicted structures from pre-trained

folding models including OmegaFold (Wu et al., 2022b) and ESMFold (Lin et al., 2022). The training target distribution consists of 200,000 frames at $5ns$ intervals of a $1ms$ molecular dynamics trajectory of the BPTI protein (Shaw et al., 2010); the inference of FOLDFLOW is tested against 20,000 unseen frames in that trajectory. FOLDFLOW can successfully model both the general set of conformations, as indicated by ICA of the dihedral angles in fig. 5b, as well as the highly flexible residues, as seen in the 2D Ramachandran plot in fig. 5a. We note that our approach can capture all the modes of distribution in contrast to AlphaFold2, which does not model the flexibility well (fig. 5c). In table 3, we observe under the 2-Wasserstein ($\mathcal{W}_2$) metric for all angles and only residue 56, FOLDFLOW with an informed prior outperforms a random prior, and FrameDiff which can only use an uninformed prior which further highlights a key advantage of FOLDFLOW over diffusion-based approaches.

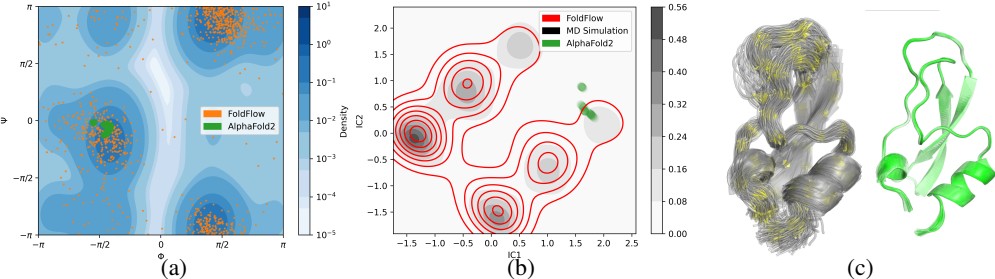

Figure 5: (a) Ramachandran plot of $\Phi$ and $\Psi$ of the most flexible residue (56) in BPTI (b) ICA of all dihedral angles of BPTI (c) 1000 BPTI conformations sampled by FOLDFLOW with $C_\alpha$ alignment highlighted in yellow and AlphaFold2 samples in green. FOLDFLOW reproduces test MD frames while AlphaFold2 samples do not.

# 6 RELATED WORK

**Protein design approaches**. The field of protein design has evolved over the course of several decades with many useful libraries (McCafferty et al., 1990; Winter et al., 1994; Romero & Arnold, 2009; Wang et al., 2021) with subfields being impacted by ML assitance (Yang et al., 2019). Sequence-based machine learning approaches resulted in multiple successful protein design cases (Madani et al., 2023; Verkuil et al., 2022; Alamdari et al., 2023; Hie et al., 2022). Structure-based biophysics approaches resulted in several drug candidates (Röthlisberger et al., 2008; Fleishman et al., 2011; Cao et al., 2020b). Moreover, diffusion-based approaches have risen in prominence (Wu et al., 2022a; Yim et al., 2023b), including significantly improved biological experiment success compared to previous SOTA (Watson et al., 2023). SE(3) diffusion has also seen applications in protein-ligand binding (Jin et al., 2023), docking (Somnath et al., 2023), as well as robotics (Brehmer et al., 2023). Lastly, FrameFlow (Yim et al., 2023a) concurrently investigates a model similar to FOLDFLOW-BASE.

**Equivariant generative models**. There have been several efforts to incorporate symmetry constraints in generative models. These include building equivariant vector fields for CNFs (Köhler et al., 2020; Katsman et al., 2021; Garcia Satorras et al., 2021; Klein et al., 2023) and finite flows using the affine coupling transform (Dinh et al., 2017; Bose & Kobyzev, 2021; Midgley et al., 2023). Applications in theoretical physics have also been impacted by equivariant flows (Boyda et al., 2020; Kanwar et al., 2020; Abbott et al., 2023). Lastly, beyond flows a new genre of models coming into prominence is based on the idea of equivariant score matching (De Bortoli et al., 2022; Brehmer et al., 2023) and diffusion models (Hoogeboom et al., 2022; Xu et al., 2022; Igashov et al., 2022).

# 7 CONCLUSION

In this paper, we tackle the problem of protein backbone generation using $\text{SE}(3)^N$-invariant generative models. In pursuit of this objective, we introduce FOLDFLOW, a family of simulation-free generative models under the flow matching framework. Within this model class, we introduce FOLDFLOW-BASE which learns deterministic dynamics over $\text{SE}(3)$, FOLDFLOW-OT which learns more stable flows using Riemannian OT. To learn stochastic dynamics, we propose FOLDFLOW-SFM which learns an SDE over $\text{SE}(3)^N$, and is motivated by learning Brownian bridges over $\text{SO}(3)$, but in a simulation-free manner. We investigate the empirical caliber of FOLDFLOW models on PDBs that contain up to 300 amino acids and find that our proposed models are competitive with RFDiffusion while significantly outperforming the current non-pretrained SOTA approach, FrameDiff-Improved, on all metrics. Finally, FOLDFLOW is more amenable for equilibrium conformation sampling which is an important subtask in protein design. Beyond generating 3D structures, a natural direction for future work is to extend FOLDFLOW to conditional generation by using target sequence and structure during training.

## ACKNOWLEDGEMENTS

AJB was supported by the Ivado Phd fellowship. KF is supported by NSERC Discovery grant (RGPIN-2019-06512), CIFAR AI Chairs, and a Samsung grant. CHL is supported by the Vanier scholarship. The authors would like to thank Clément Bonet and Gauthier Gidel for fruitful discussions on Riemannian geometry and optimal transport. We also thank Riashat Islam for providing feedback on early versions of this work. We thank Alexander Stein and the entire DreamFold team for providing a vibrant workspace that enabled this research. The authors would like to thank the restaurant Le Don Donburi for keeping our stomachs full and sustaining this research. Finally, the authors would like to acknowledge Anyscale and Google GCP for providing computational resources for the protein experiments.

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

# A  THEORETICAL PRELIMINARIES

## A.1  SO(3) LIE GROUP

The Special Orthogonal group in 3 dimensions, SO(3) consists of the 3D rotation matrices:

$$\text{SO}(3) = \left\{ r \in \mathbb{R}^{3\times3} : r^\top r = rr^\top = I, \det r = 1 \right\} \tag{11}$$

It is a matrix Lie group with the lie algebra given by:

$$\mathfrak{so}(3) = \left\{ \mathfrak{r} \in \mathbb{R}^{3\times3} : \mathfrak{r}^\top = -\mathfrak{r} \right\} \tag{12}$$

**Parametrizations of** SO(3). The skew-symmetric matrices $\mathfrak{r} \in \text{SO}(3)$ can be uniquely identified with a vector $\boldsymbol{\omega} \in \mathbb{R}^3$ such that $\forall \mathbf{v} \in \mathbb{R}^3$, $\mathfrak{r}\mathbf{v} = \boldsymbol{\omega} \times \mathbf{v}$, where $\times$ indicates the cross product. This vector is known as the *rotation vector*. The magnitude of this vector, $\omega = ||\boldsymbol{\omega}||$ is the angle of rotation its direction, $e_{\boldsymbol{\omega}} = \frac{\boldsymbol{\omega}}{||\boldsymbol{\omega}||}$ is the axis of rotation.

Mapping the $\mathbb{R}^3$ vector to the skew-symmetric matrix is known as the *hat* operation, $(\hat{\cdot})$.

Another parametrization of SO(3) is with *Euler angles*, described using three angles $(\phi, \theta, \psi)$. A common convention is to use the $x$-convention, where the rotation is given by: a rotation about the $z$-axis by $\phi$, a second rotation about the former $x$-axis by $\theta$, and a last one about the former $z$-axis by $\psi$.

**Metric on** SO(3). First, we recall that a metric is a bilinear function $\langle \cdot, \cdot \rangle : \mathbb{R}^n \times \mathbb{R}^n \to \mathbb{R}$ that is both symmetric and positive definite. Additionally, we recall that a quadratic form on a manifold $\mathcal{M}$ is a bilinear map $\mathcal{T}_x\mathcal{M} \times \mathcal{T}_x\mathcal{M} \to \mathbb{R}$ that is smooth and symmetric. A positive-definite quadratic form is, therefore, a metric. Let us consider the following symmetric positive definite quadratic form defined:

$$Q = \begin{pmatrix} A & B^\top \\ B & C \end{pmatrix}. \tag{13}$$

A canonical choice for the metric of SO(3) is obtained by taking $Q = 1/2I$, resulting in a bi-invariant metric on SO(3). Therefore, the metric is given by:

$$\langle \mathfrak{r}_1, \mathfrak{r}_2 \rangle_{\text{SO}(3)} = \text{tr}(\mathfrak{r}_1^\top Q \mathfrak{r}_2) = \frac{1}{2}\text{tr}(\mathfrak{r}_1^\top \mathfrak{r}_2) \tag{14}$$

Note that the inner product on Lie groups consumes elements of the Lie algebra and, because the left action is transitive, this inner product is well-defined for all tangent spaces of the group elements.

The distance induced by this metric is given by:

$$d_{\text{SO}(3)}(r_1, r_2) = \| \log (r_1^\top r_2) \|_F \tag{15}$$

for $r_1, r_2 \in \text{SO}(3)$ and where the Frobenius matrix norm is used.

**The exponential and logarithmic maps on** SO(3). Generally speaking, the *exponential* and *logarithmic* maps of a Lie group $G$ relate the elements in the group to the lie algebra, $\mathfrak{G}$. In the case of matrix Lie groups, these coincide with the *matrix exponential*:

$$g = \exp(\mathfrak{g}) = \sum_{n=0}^{N} \frac{1}{n!}\mathfrak{g}^n \tag{16}$$

and the *matrix logarithm*:

$$\mathfrak{g} = \log(g) = \sum_{n=1}^{N} \frac{(-1)^{n-1}}{n}(g-I)^n \tag{17}$$

For SO(3), since the elements of the lie algebra are skew-symmetric matrices, eq. (16) for the matrix exponential can be simplified significantly to obtain a closed-form, known as Rodrigues formula. Given $\boldsymbol{\omega}$ a rotation vector, and $\hat{\boldsymbol{\omega}} \in \mathfrak{so}(3)$, the corresponding element of the lie group, $r \in \text{SO}(3)$ is given by:

$$r = \exp\hat{\boldsymbol{\omega}} = \cos(\omega)I + \sin(\omega)e_{\boldsymbol{\omega}} + (1 - \cos(\omega))e_{\boldsymbol{\omega}}e_{\boldsymbol{\omega}}^\top \tag{18}$$

where $\omega$ and $e_{\boldsymbol{\omega}}$ are the angle and axis of rotation for $\boldsymbol{\omega}$.

Similarly, the matrix logarithm can be expressed using the rotation angle:

$$\log(r) = \begin{cases} \frac{\omega}{2\sin(\omega)}(r - r^\top) & \text{if } r \neq I, \\ 0 & \text{if } r = I. \end{cases} \tag{19}$$

## A.2 SE(3) LIE GROUP

The special Euclidean group, SE(3) is used to represent rigid body transformations in 3 dimensions:

$$\mathrm{SE}(3) = \left\{ \begin{pmatrix} r & s \\ 0 & 1 \end{pmatrix} : r \in \mathrm{SO}(3), s \in (\mathbb{R}^3, +) \right\} \tag{20}$$

Represented by this $4 \times 4$ matrix and with the group operation defined by matrix multiplication, this group can be seen as a subgroup of the general linear group $\mathrm{GL}(4, \mathbb{R})$. The lie algebra of the group $\mathfrak{se}(3)$ is given by:

$$\mathfrak{se}(3) = \left\{ \mathfrak{x} = \begin{pmatrix} \mathfrak{r} & s \\ 0 & 0 \end{pmatrix} : \mathfrak{r} \in \mathfrak{so}(3), s \in \mathbb{R}^3 \right\} \tag{21}$$

Note that the tangent space of $\mathbb{R}^3$ is isomorphic to the space itself so we can simply use the notation $s$ instead of $\mathfrak{s}$. This lie algebra is isomorphic to $\mathbb{R}^6$ using the map: $\mathfrak{x} \mapsto (\boldsymbol{\omega}, s)$, where we have identified the skew-symmetric matrix $\mathfrak{r} \in \mathfrak{so}(3)$ with its axis-angle representation, $\boldsymbol{\omega} \in \mathbb{R}^3$. As the group of translations, $(\mathbb{R}^3, +)$ is a normal subgroup of SE(3), the group can be understood as a semi-direct product: $\mathrm{SE}(3) = \mathrm{SO}(3) \ltimes (\mathbb{R}^3, +)$.

**Metric on** SE(3). Although there are many possible choices for metrics on SE(3), none of them are bi-invariant. Instead, one can choose to build a left-invariant or right-invariant metric. A simple choice for the quadratic form $Q$ from eq. (13) is setting the matrices $A = C = I_3$ and $B = 0$ (Park & Brockett, 1994), which gives:

$$Q = \begin{pmatrix} I_3 & 0 \\ 0 & I_3 \end{pmatrix}. \tag{22}$$

Using this metric we can define an inner product on SE(3) as $\langle \mathfrak{x}_1, \mathfrak{x}_2 \rangle_{\mathrm{SE}(3)} = \mathrm{tr}(\mathfrak{x}_1^\top Q \mathfrak{x}_2)$, where tr is the trace operation. Writing out the inner product explicitly for $\mathfrak{x}_1, \mathfrak{x}_2 \in \mathfrak{se}(3)$ we get,

$$\mathrm{tr}(\mathfrak{x}_1^\top Q \mathfrak{x}_2) = \mathrm{tr} \begin{pmatrix} \mathfrak{r}_1^\top \mathfrak{r}_2 & \mathfrak{r}_1^\top \mathfrak{s}_2 \\ \mathfrak{s}_1^\top \mathfrak{r}_2 & \mathfrak{s}_1^\top \mathfrak{s}_2 \end{pmatrix}. \tag{23}$$

Thus, we have $\mathrm{tr}(\mathfrak{x}_1^\top Q \mathfrak{x}_2) = \mathfrak{r}_1^\top \mathfrak{r}_2 + s^\top s_2$. Therefore, the metric on SE(3) decomposes into the metric on SO(3) and $\mathbb{R}^3$:

$$\langle \mathfrak{x}_1, \mathfrak{x}_2 \rangle_{\mathrm{SE}(3)} = \langle \mathfrak{r}_1, \mathfrak{r}_2 \rangle_{\mathrm{SO}(3)} + \langle s_1, s_2 \rangle_{\mathbb{R}^3} \tag{24}$$

This means that we can obtain the geodesics on SE(3) from the geodesics on the product manifold $\mathrm{SO}(3) \times \mathbb{R}^3$:

$$d_{\mathrm{SE}(3)}(x_1, x_2) = \sqrt{d_{\mathrm{SO}(3)}(r_1, r_2)^2 + d_{\mathbb{R}^3}(s_1, s_2)^2} \tag{25}$$

where $x_1 = (r_1, s_1), x_2 = (r_2, s_2) \in \mathrm{SE}(3)$, $d_{\mathrm{SO}(3)}$ is defined in eq. (15) and $d_{\mathbb{R}^3}$ is the usual Euclidean distance.

## A.3 THE ISOTROPIC GAUSSIAN DISTRIBUTION ON SO(3)

$\mathcal{IG}_{\mathrm{SO}(3)}$ **density**. The isotropic Gaussian distribution on SO(3) is parametrized by a mean, $r \in \mathrm{SO}(3)$ and a concentration parameter, $\epsilon \in \mathbb{R}$. It can be parametrized in axis-angle, where the axis of rotation is sampled uniformly and the angle of rotation $\omega$ has probability density function (pdf) given by:

$$f(\omega_x, \epsilon) = \sum_{l=0}^{\infty} (2l + 1) e^{-l(l+1)\epsilon} \frac{\sin\left((l + 1/2)\omega_x\right)}{\sin(\omega_x/2)} \tag{26}$$

Although this expression contains an infinite sum, Matthies et al. (1900) has shown that for $\epsilon \leq 1$, it can be approximated by a closed-form equation:

$$f(\omega_x, \epsilon) = \sqrt{\pi} \epsilon^{-3/2} e^{\frac{\epsilon - \omega^2/\epsilon}{4}} \frac{\left(\omega - e^{-\pi^2/\epsilon}\left((\omega - 2\pi)e^{\pi\omega/\epsilon} + (\omega + 2\pi)e^{-\pi\omega/\epsilon}\right)\right)}{2\sin\left(\frac{\omega}{2}\right)} \tag{27}$$

**Sampling from $\mathcal{IG}_{\mathrm{SO}(3)}$.** Sampling from $\mathcal{IG}_{\mathrm{SO}(3)}$ is done following Leach et al. (2022). The angle of rotation is obtained by inverse transform sampling, where the cumulative density function is approximated using the pdf above, scaled by uniform density on SO(3) with density $f(\omega) = \frac{1-\cos\omega}{\pi}$; the axis is sampled uniformly from $\mathbb{S}^2$. We note that the closed-form approximation of eq. (27) makes the computation of the cdf, and hence the sampling process very efficient.

## B  FLOW MATCHING IN $\mathbb{R}^d$

To perform FOLDFLOW on SE(3), we consider two different flows. One on SO(3) that we described in the main paper and another one on $\mathbb{R}^9$ that we describe depending on the consider FOLDFLOW method.

Riemannian Flow Matching is a generalization of Flow Matching on Riemannian manifold. Therefore, the setting as well as the main ideas are similar and are straightforward to adapt to the Euclidean case. This means that the objective is also to regress a conditional vector field built from conditional probability paths. In this section, we described the conditional probability paths and conditional vector fields that were used respectively by Lipman et al. (2022) and Tong et al. (2023b).

The main difference is that the conditional probability path is now a Gaussian conditioned on a latent variable $z \sim q(z)$ with variance $\sigma_t$, $\rho_t(s) = \mathcal{N}(s|z, \sigma_t)$. The conditional vector field has a closed form derived from the following Theorem:

**Theorem 1** (Theorem 3 of Lipman et al. (2022)). *The unique vector field whose integration map satisfies $\rho_t(s) = \mu_t + \sigma_t s$ has the form*

$$u_t(s) = \frac{\sigma_t'}{\sigma_t}(s - \mu_t) + \mu_t', \tag{28}$$

We now describe the Flow Matching method Lipman et al. (2022) and OT-CFM Tong et al. (2023b;a); Pooladian et al. (2023b).

**Flow Matching.** In the context of data living in the Euclidean space $\mathbb{R}^d$. Identifying the condition $z$ with a single datapoint $z := s_1$, and choosing a smoothing constant $\sigma > 0$, one sets

$$p_t(s|z) = \mathcal{N}(s \mid ts_1, (t\sigma - t + 1)^2), \tag{29}$$

$$u_t(s|z) = \frac{s_1 - (1-\sigma)s}{1 - (1-\sigma)t}, \tag{30}$$

which is a probability path from the standard normal distribution $(p_0(x|z) = \mathcal{N}(x; 0, 1))$ to a Gaussian distribution centered at $x_1$ with standard deviation $\sigma$ $(p_1(x|z) = \mathcal{N}(x; x_1, \sigma^2))$. If one sets $q(z) = q(x_1)$ to be the uniform distribution over the training dataset, the objective introduced by Lipman et al. (2022) is equivalent to the CFM objective (1) for this conditional probability path.

**OT-Conditional Flow Matching (Tong et al., 2023b).** As explained in the main paper, the probability path used in FM is not the optimal transport probability paths between the distributions $\rho_0$ and $\rho_1$. Therefore, we want to get straighter flows for faster inference and more stable training. To achieve that, we leverage the optimal transport theory and want the probability path to be the Euclidean McCann interpolants defined as $\rho_t = t\Psi(s_0) + (1-t)s_0$. However, as the map $\Psi$ is intractable in practice, we rely on the Brenier theorem which makes a connection between the map $\Psi$ and the optimal transport plan $\pi$. Therefore we set the mean of Gaussian conditional probability path as $\mu_t = ts_1 + (1-t)s_0$ and the latent distribution $q(s_0, s_1) = \pi(s_0, s_1)$.

$$p_t(s|s_0, s_1) = \mathcal{N}(s \mid ts_1 + (1-t)s_0, \sigma^2), \tag{31}$$

$$p_t(s) = \int \mathcal{N}(s \mid ts_1 + (1-t)s_0, \sigma^2)\pi(s_0, s_1)ds_0 ds_1, \tag{32}$$

$$u_t(s|z) = s_1 - s_0. \tag{33}$$

In the case of the Euclidean space, the FM loss is equal to $\mathcal{L}_{\mathrm{FOLDFLOW}-\mathbb{R}^3} = \|v_\theta(t, s) - u_t(s|z)\|$. This can be simplified to down to $\mathcal{L}_{\mathrm{FOLDFLOW}-\mathbb{R}^3} = \|v_\theta(t, s) - (s_1 - s_0)\|$. This method is the main inspiration to develop FOLDFLOW-OT.

## C    Riemannian Optimal transport

**Optimal transport in generative models**. OT has been used in generative models for several approaches. For GANs, it was used as a loss function (Genevay et al., 2018; Fatras et al., 2021b; Salimans et al., 2018; Arjovsky et al., 2017). More recently, it was used to speeding up training and inference for continuous normalizing flows (Finlay et al., 2020; Liu et al., 2023b; Lipman et al., 2022; Tong et al., 2020; 2023b;a), Schrödinger bridge models (Shi et al., 2023; Liu et al., 2023a; De Bortoli et al., 2021). In this section, we recall its basic definition over a Riemannian manifold. Then we discuss its empirical computation and we finish this section by proving Proposition 1.

### C.1    Background on Riemannian Optimal transport

Optimal transport on Riemannian manifold was first studied in the seminal work of McCann (2001) and we refer to Villani (2003; 2008) for a review of all results. Recently, optimal transport has also drawn attention from the machine learning community, and we now give a longer introduction on this topic.

The (static) Kantorovich optimal transport problem seeks a mapping from one measure to another that minimizes a displacement cost. Formally, we define the 2-Wasserstein distance between distributions $\rho_0$ and $\rho_1$ on $\mathcal{M}$ with respect to the cost $c(x,y) = \frac{1}{2}d(x,y)^2$ as:

$$W(\rho_0, \rho_1)_2^2 = \inf_{\pi \in \Pi(\rho_0, \rho_1)} \int_{\mathcal{M}^2} c(x,y) \, d\pi(x,y), \tag{34}$$

where $\Pi(\rho_0, \rho_1)$ denotes the set of all joint probability measures on $\mathcal{M} \times \mathcal{M}$ whose marginals are $\rho_0$ and $\rho_1$. To compute the optimal transport plan, we rely on the POT library Flamary et al. (2021). This problem is a relaxation of the well-known Monge formulation described in the main paper and that we recall now for the sake of readability.

The Monge optimal transport problem is defined as

$$\mathrm{OT}(\rho_0, \rho_1) = \inf_{\Psi : \Psi_\# \rho_0 = \rho_1} \int_{\mathcal{M}} c(x, \Psi(x)) \, d\rho_0(x). \tag{35}$$

When $\mathcal{M}$ is a smooth compact manifold with no boundary and $\rho_0$ has a density, (McCann, 2001, Proposition 9) shows that the map $T$ exists and is unique. This is an extension to Riemannian manifold of the well-known Brenier Theorem (Brenier, 1991). The optimal transport map $\Psi$ and the McCann interpolation have then the following form:

$$\Psi(x) = \exp_x(-\nabla\phi(x)), \qquad \Psi_t(x) = \exp_x(-t\nabla\phi(x)), \tag{36}$$

where $\phi$ is a $c$-concave function. Furthermore, we have that the optimal transport plan is supported on the graph of the Monge map, *i.e.,* $\pi = (\mathrm{id}, \Psi)_\# \rho_0$. Therefore, knowing the transport plan leads to the Monge map. The connection between the two formulations for $\mathrm{SE}(3)_0^N$ is explicitly stated in Proposition 1 which is proved in §C.2. However, we first discuss the computation of the OT plan $\pi$.

**Minibatch OT approximation.**    For empirical distributions, the Kantorovich problem is a linear program and can be efficiently solved with the simplex algorithm. We refer to (Peyré & Cuturi, 2019, Chapter 3) for a review on how to solve the Kantorovich problem. However, when we deal with large datasets, computing and storing the transport plan $\pi$ for Optimal Transport (OT) can be challenging due to its cubic time and quadratic memory complexity with respect to the number of samples. To address this, a minibatch OT approximation is often employed. While this approach introduces some error compared to the exact OT solution (Fatras et al., 2020), it has been proven effective in various applications such as domain adaptation and generative modeling (Damodaran et al., 2018; Genevay et al., 2018). Specifically, during training, for each source and target minibatch, pairs of points are sampled from the optimal transport plan computed between the pair $(x, y) \sim \pi_{\mathrm{batch}}$. We empirically show that the batch size can be small compared to the full dataset size and still give a good performance, which aligns with prior studies (Fatras et al., 2021b;a). This strategy is also at the heart of the OT-CFM methods (Tong et al., 2023b;a; Pooladian et al., 2023b).

## C.2 PROOF OF PROPOSITION 1

We recall the proposition statement here for convenience and then prove it below.

> **Proposition 1.** *Let us consider* $\mathrm{SE}(3)_0^N$ *with the product distance* $d_{\mathrm{SE}(3)_0^N}$ *and two compactly supported probability distributions* $\rho_0, \rho_1 \in \mathbb{P}(\mathrm{SE}(3)_0^N)$. *In addition, suppose that* $\rho_0$ *is absolutely continuous with respect to Riemannian volume form* (i.e., $\rho_0 \ll dx$). *Then for the distance* $c = \frac{1}{2}d_{\mathrm{SE}(3)_0^N}^2$, *the Kantorovich and Monge problems admit a unique solution that is connected as follows* $\pi = (id \times \Psi)_\# \rho_0$, *where* $\Psi$ *is almost uniquely determined everywhere* $\rho_0$. *Furthermore, we have that* $\Psi(x) = \exp_x(\nabla\phi(x))$ *for some* $d_{\mathrm{SE}(3)_0^N}^2$*-concave function* $\phi$.

*Proof.* The manifold $\mathrm{SE}(3)$ is a connected, complete, ($\mathcal{C}^\infty$) smooth manifold without boundary. $\mathrm{SE}(3)^N$ (equipped with the usual product distance) is a finite Cartesian product of connected, complete, smooth manifolds without boundary and therefore it is itself a connected, complete, smooth manifold without boundary. We only need to check these assumptions are also satisfied by $\mathrm{SE}(3)_0^N$. We do so by noting that $\mathrm{SE}(3)_0^N$ can be written as $N-1$ copies of $\mathrm{SE}(3)$ where the $\mathbb{R}^3$ component is mean subtracted—i.e. $s^c = s - 1/N\sum_{i=1}^N s^i$, and the final $N$th element in the product is the mean, $1/N\sum_{i=1}^N s_i$. Certainly, the first $N-1$ components satisfy connectedness, and completeness, and are manifolds that are smooth without boundary. Furthermore, the disintegration of measures on $\mathrm{SE}(3)_0^N$ (Proposition 3.5 (Yim et al., 2023b)) allows us to define a measure $\bar{\mu}$ proportional to $\mathbb{R}^3$ for the final component $N$th component. Therefore, by our assumptions on the measures $\rho_0, \rho_1$, we can apply the following Theorem from Villani (2003) to get the desired results.

**Theorem 2** (Theorem 2.47, Villani (2003)). *Let* $\mathcal{M}$ *be a connected, complete and smooth* ($\mathcal{C}^3$) *Riemannian manifold without boundary, equipped with its standard volume measure. Let* $\rho_0, \rho_1$ *be two compactly supported distributions and set the ground cost* $c(x, y) = \frac{1}{2}d(x, y)^2$ *with* $d$ *the geodesic distance on* $\mathcal{M}$. *Further, assume that* $\rho_0$ *is absolutely continuous with respect to the volume measure on* $\mathcal{M}$. *Then the Kantorovich and Monge problems admit a unique solution that is connected as follows* $\pi = (id \times \Psi)_\# \rho_0$, *where* $\Psi$ *is almost uniquely determined everywhere* $\rho_0$. *Furthermore we have that* $\Psi(x) = \exp_x(\nabla\phi(x))$ *for some* $d^2$*-concave function* $\phi$.

$\square$

# D   STOCHASTIC RIEMANNIAN FLOW MATCHING

## D.1   BROWNIAN BRIDGE ON $\mathrm{SO}(3)$

We follow the presentation in Jensen et al. (2022) to define the Brownian bridge on a Lie group $G$ endowed with a metric. We note that $\log$ is the inverse of the Riemannian exponential map. However, if the metric is bi-invariant, which is the case for $\mathrm{SO}(3)$, it coincides with the Lie group logarithm. We can simulate a bridge on $G$ via the guided diffusion SDE (using $\circ$ for the Stratonovich integral), for a process conditioned to reach $v$ at $t = 1$.

$$\mathrm{dR_t} = -\frac{1}{2}\mathrm{V_0(R_t)}\,\mathrm{dt} + \mathrm{V_i(R_t)} \circ \left(\mathrm{dB_t^i} - \frac{\log_{\mathrm{R_t}}(\mathrm{v})^i}{1-\mathrm{t}}\,\mathrm{dt}\right) \quad \mathrm{R_0} = \mathrm{r_0}, \tag{37}$$

where $V_i(xr = (dL_r)_e v_i$ with $\{v_1, \ldots, v_d\}$ an orthonormal basis of $T_e G$, and where $\mathrm{B_t}$ is a Brownian motion on $G$. On $\mathrm{SO}(3)$, since the metric is bi-invariant, we have $V_0 = 0$. In this work, we model the guided bridge with a diffusion that does not depend on the process $\mathrm{R_t}$. In this case, the Stratonovich and Itô formulations are the same, yielding the reversed process defined in Eq. 7.

## D.2   SIMULATION-FREE APPROXIMATION OF BROWNIAN BRIDGES ON $\mathrm{SO}(3)$

We now numerically investigate the fidelity of our simulation-free SDE which is employed in FOLDFLOW-SFM in relation to the guided drift SDE in eq. (8). In fig. 6 we plot the mean and the standard deviation (over 1024 data points) of the distribution of the $\mathrm{SO}(3)$-norm along the trajectory against time, for three different values of the diffusion coefficient, $\gamma$. We find the true simulated Brownian bridge (bold black line) is in close proximity to the simulation-free FOLDFLOW-SFM SDE (red dotted lines). We further note that this holds for the entire trajectory and leads to overlapping shaded regions that correspond to the standard deviation of the norm. This result adds empirical

substantiation to using the FOLDFLOW-SFM as a drop-in and fast approximation to the Brownian bridge SDE on SO(3).

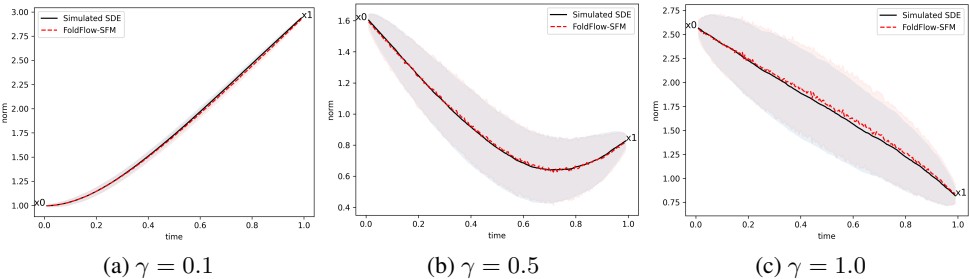

(a) $\gamma = 0.1$      (b) $\gamma = 0.5$      (c) $\gamma = 1.0$

Figure 6: Numerical comparison between the simulation-free of the SDE in FOLDFLOW-SFM vs. simulated Brownian bridge on SO(3), for different values of the diffusion coefficient, $\gamma$.

### D.3 PROOF OF PROPOSITION 2

**Proposition 2.** *Given* $\rho_t(x) > 0$, $\forall x \in \mathrm{SE}(3)_0^N$, *the conditional and unconditional* FOLDFLOW-SFM *losses have equal gradients* w.r.t. $\theta$: $\nabla_\theta \mathcal{L}_{\text{USFM}}(\theta) = \nabla_\theta \mathcal{L}_{\text{SFM}}(\theta)$.

*Proof.* Let $u_t = \mathbb{E}_{\rho(z)} \left[ \frac{\rho_t(x|z)}{\rho_t(x)} u_t(x|z) \right]$, for $x \in \mathrm{SE}(3)_0^N$. We claim that:

$$\nabla_\theta \mathbb{E}_{z \sim \rho(z), x \sim \rho_t(x|z)} \left[ ||v_\theta(t, x) - u_t(x|z)||^2_{\mathrm{SE}(3)_0^N} \right] = \nabla_\theta \mathbb{E}_{x \sim \rho_t(x)} \left[ ||v_\theta(t, x) - u_t(x)||^2_{\mathrm{SE}(3)_0^N} \right] \tag{38}$$

From disintegration of measures (Pollard (2002) and Proposition 3.5 form Yim et al. (2023b)), we know the probabilities $\rho_t(x) \propto \rho_t(r^1) \cdots \rho_t(r^N) \rho_t(s^1) \cdots \rho_t(s^N)$, and similar for the conditional probability $\rho_t(x|z)$. Given that by eq. (24), the metric on SE(3) also factorizes into metric on SO(3) and $\mathbb{R}^3$, it suffices to prove the claim for SO(3) and $\mathbb{R}^3$. The claim can therefore be stated as follows, where we have written $r^i$ as $r$ and $s^i$ as $s$ for conciseness.

$$\nabla_\theta \mathbb{E}_{z \sim \rho(z), x \sim \rho_t(r|z_r)} \left[ ||v_\theta(t, r) - u_t(r|z_r)||^2_{\mathrm{SO}(3)} \right] = \nabla_\theta \mathbb{E}_{r \sim \rho_t(r)} \left[ ||v_\theta(t, r) - u_t(r)||^2_{\mathrm{SO}(3)} \right] \tag{39}$$

$$\nabla_\theta \mathbb{E}_{z \sim \rho(z), s \sim \rho_t(s|z_s)} \left[ ||v_\theta(t, s) - u_t(s|z_s)||^2_{\mathbb{R}^3} \right] = \nabla_\theta \mathbb{E}_{s \sim \rho_t(s)} \left[ ||v_\theta(t, s) - u_t(s)||^2_{\mathbb{R}^3} \right] \tag{40}$$

The proof of this claim follows a similar structure to Chen & Lipman (2023). We proceed by proving eq. (40). Dropping the distributions for conciseness, we have:

$$\nabla_\theta \left( \mathbb{E}_{z_r, r} [||v_\theta(t, r) - u_t(r|z_r)||^2] - \mathbb{E}_{z_r, r} [||v_\theta(t, r) - u_t(r)||^2] \right)$$
$$= \nabla_\theta \left( -2 \mathbb{E}_{z_r, r} \langle v_\theta(t, r), u_t(r|z_r) \rangle_{\mathrm{SO}(3)} - 2 \mathbb{E}_r \langle v_\theta(t, r), u_t(r) \rangle_{\mathrm{SO}(3)} \right) \tag{41}$$

Now:

$$\begin{aligned}
\mathbb{E}_r \langle v_\theta(t, r), u_t(r) \rangle &= \int_0^1 \int_{\mathrm{SO}(3)} \langle v_\theta(t, r), u_t(r) \rangle_{\mathrm{SO}(3)} \rho_t(r) d\mathrm{vol}_r \\
&= \int_0^1 \left\langle v_\theta(t, r) \int_{\mathrm{SO}(3)} \frac{\rho_t(r|z_r)}{\rho_t(r)} u_t(r|z_r) \rho(z_r) d\mathrm{vol}_{z_r} \right\rangle \rho_t(r) dr \\
&= \int_0^1 \int_{\mathrm{SO}(3)} \langle v_\theta(t, r), u_t(r|r_z) \rangle \rho_t(r|r_z) \rho(r_z) d\mathrm{vol}_r d\mathrm{vol}_{z_r} \\
&= \mathbb{E}_{r, z_r} \langle v_\theta(t, r) u_t(r|z_r) \rangle
\end{aligned} \tag{42}$$

The proof for $\mathbb{R}^3$ directly follows Theorem 3.2 from Tong et al. (2023a).

# E    EXTENDED FIGURE INFORMATION

## E.1    FIGURE 1

depicts the probability paths of FOLDFLOW-BASE, FOLDFLOW-OT, and FOLDFLOW-SFM projected onto $\mathbb{S}^2$. Where FOLDFLOW-BASE paths may cross, FOLDFLOW-OT conditional paths do not cross reducing the variance in the objective stabilizing training as studied in Pooladian et al. (2023b); Tong et al. (2023b). FOLDFLOW-SFM adds in stochasticity which improves novelty in our protein generation task. Figure 1 also contains a table summarizing the differences between methods. Showing whether or not they can map from a general source distribution, can perform optimal transport under some conditions, are stochastic or deterministic, and do not require calculation of the score. We note that there is a $^*$ for FOLDFLOW-SFM performing OT, as it only achieves OT when noise goes to zero and it recovers FOLDFLOW-OT However, this bias may still be helpful in reducing the variance of the objective function even if OT is not achieved.

# F    SO(3) TOY EXPERIMENT

## F.1    TOY MODEL PARAMETERIZATION

For the vector-field parametrization, the goal is to create a function that by construction lies on the tangent space of the manifold. For the toy experiments, this is done by using a 3-layer MLP, and projecting the output of the network to the tangent space of the input. That is, similar to Chen & Lipman (2023), we have:

$$u_\theta(t, r) = \Pi_r \text{MLP}(t, r),    \tag{43}$$

where $\Pi_r(M)$ projects a $3 \times 3$ matrix onto $\mathcal{T}_r \text{SO}(3)$. This operation essentially computes the skew-symmetric component of $M$, given by $\frac{M - M^\top}{2}$ and parallel transports it to the tangent space of $r$ using left matrix multiplication which is the group operation on $\text{SO}(3)$.

## F.2    ADDITIONAL RESULTS FOR SO(3) TOY EXPERIMENTS

In this section, we present the qualitative results of our toy experiments. In fig. 7, we can see that all the three models, FOLDFLOW-BASE, FOLDFLOW-OT FOLDFLOW-SFM learn to correctly model the modes of the ground-truth distribution with a slight model shrinkage in the FOLDFLOW-BASE.

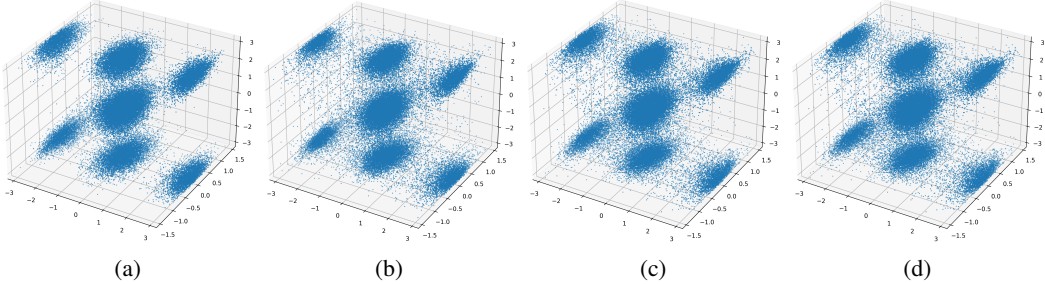

|  (a)  |  (b)  |  (c)  |  (d)  |

Figure 7: (a) Data distribution (b) FOLDFLOW-BASE (c) FOLDFLOW-OT (d) FOLDFLOW-SFM. The data is visualized using the Euler-angle representation of the rotation matrices.

# G    ADDITIONAL RESULTS AND ANALYSIS FOR THE PROTEIN EXPERIMENTS

## G.1    PROTEIN BACKBONE GENERATION EXPERIMENT ADDITIONAL RESULTS

**Empirical investigation of rotation norms for Inference Annealing**.

In fig. 10, we show five proteins generated by FOLDFLOW-SFM from each backbone length. Here we show the generated structure in green and the best ESM-refolded structure out of eight sequences generated using ProteinMPNN. We can see that FOLDFLOW-SFM generates diverse folds that refold with diversity in secondary structure and overall 3D conformation.

In fig. 9 we compare the performance of models on designability, diversity, and novelty tasks for different backbone lengths. In particular, we can see that FOLDFLOW closes the gap between models

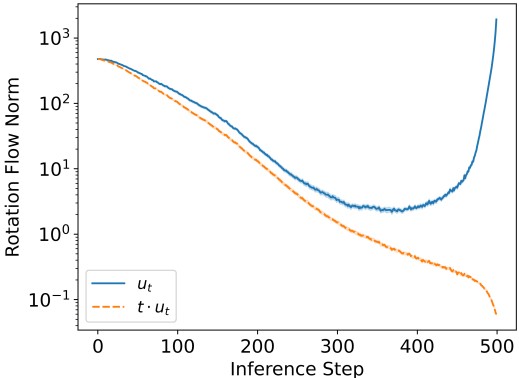

Figure 8: Norm of the rotation flow with and without $t$ scaling.

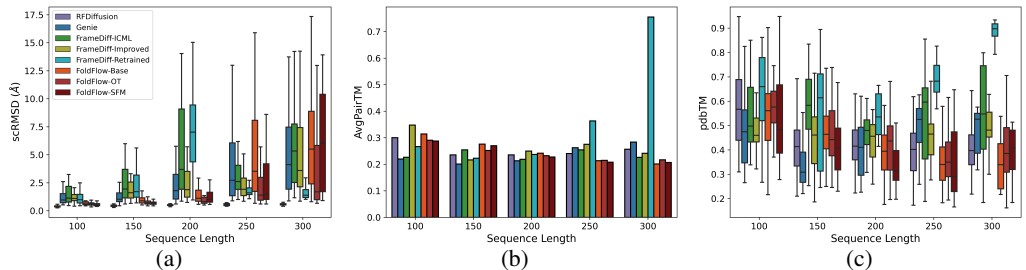

Figure 9: (a) Designability as quantified by scRMSD (lower is better), (b) Diversity as quantified by average pairwise TMScore (lower is better), and (c) Novelty of proteins as quantified by maximum TMScore to PDB (lower is better), designed across lengths for FOLDFLOW models and previous state of the art models.

without pretraining (Genie, FrameDiff, FOLDFLOW) and RFDiffusion in terms of designability, particularly on shorter sequences ($\leq 200$).

We note there is a trade-off between designability and diversity/novelty, both at the short sequence lengths and as sequence length increases. For longer sequences (250, 300), while FOLDFLOW models are comparable in terms of designability, they generate significantly more diverse and novel structures as compared to all other models, even RFDiffusion (although RFDiffusion still generates significantly more designable proteins at these lengths.

### G.1.1 TIMING COMPARISON IN TABLE 2 AND TABLE 4

In Table 2 we compare the number of steps per second for each model, where a step corresponds to a forward and backwards pass on the effective batch size as defined in Equation (46) on a single GPU. Here we find that FOLDFLOW is over 2x faster than FrameDiff per step. This drastic improvement is due to a number of optimizations, with the largest being that we can avoid the costly $\mathcal{IG}_{\mathrm{SO}(3)}$ score computation which is necessary for their method.

We train our model in Pytorch using distributed data-parallel (DDP) across four NVIDIA A100-80GB GPUs for roughly 2.5 days. We note that this is substantially less than comparable models (table 4). RFDiffusion requires the use of pre-trained weights from RosettaFold which trained for 4 weeks on 64 V100 GPUs (Watson et al., 2023).

### G.2 EQUILIBRIUM CONFORMATION GENERATION EXPERIMENT

As described in Section 5.3 proteins take on many different physical conformations in the real world. These conformations dictate many important attributes of a protein's behaviour, e.g., how one protein might bind to another. As a protein's conformations generally do not deviate greatly from one another, a desirable approach would be to start from a noised version of a known conformation of the protein to generate another conformation. We hypothesize that the flows required to do this are easier to learn than starting from an uninformed source distribution. We find FOLDFLOW is an ideal candidate for this setting, and we show its efficacy in fig. 5.

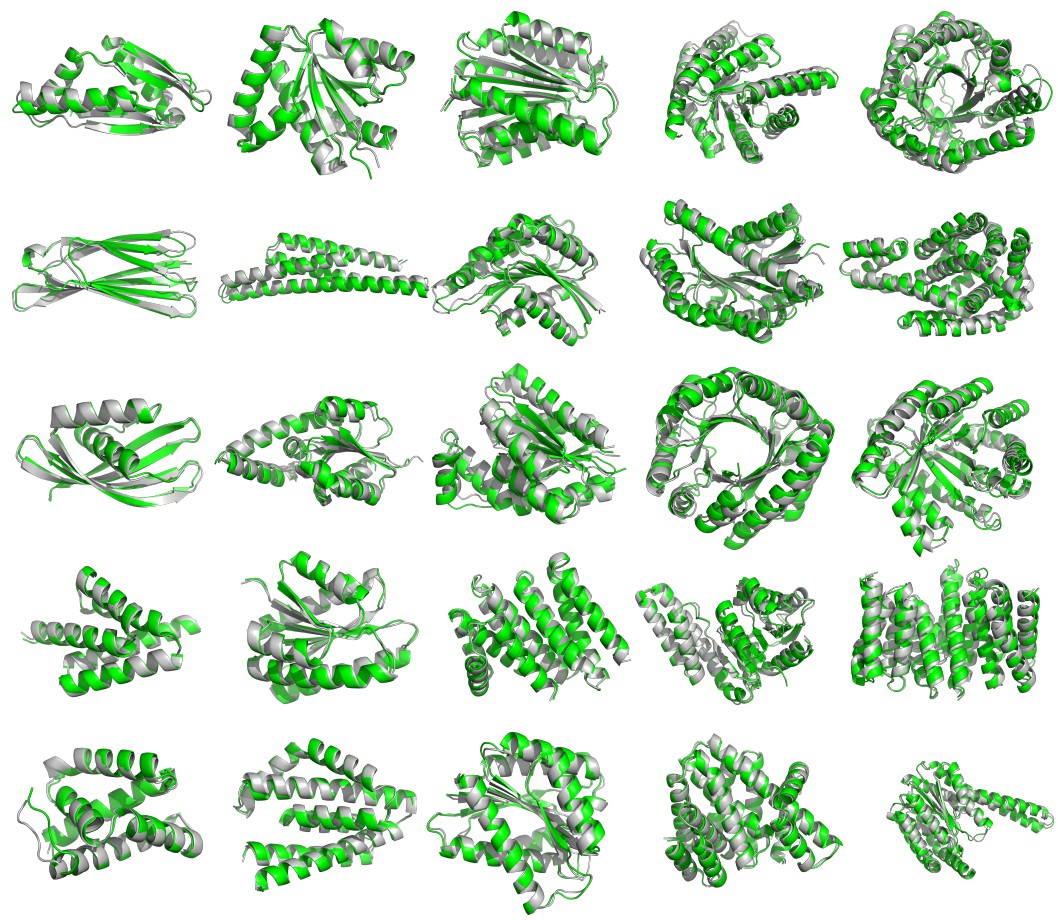

Figure 10: More FOLDFLOW-SFM generated structures in green compared to ProteinMPNN –>
ESMFold refolded structures in grey. 5 samples all with RMSD < 2Å for lengths 100, 150, 200, 250,
300 from left to right. FOLDFLOW-SFM generates designable diverse proteins.

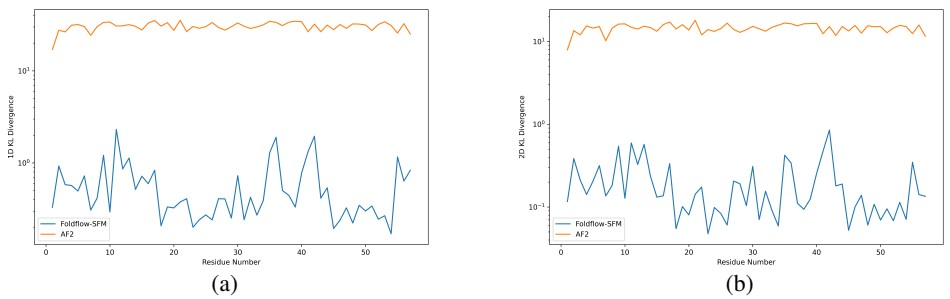

Figure 11: KL divergence per residue of the 2D dihedral angle ($\Phi$ and $\Psi$) distributions between the samples
from FOLDFLOW and test MD frames (blue) and AlphaFold 2 and the test MD frames (orange).

Table 4: Training resources for protein generation models.

| Model | Training time | Optimization Steps | #gpus | Distributed Training |
|---|---|---|---|---|
| RFDiffusion | 28 + 3 days | — | 64 + 8 | — |
| RFDiffusion w/o pretraining | 3 days | — | 8 | — |
| Genie (SwissProt) | ∼8 days | ∼800k | 6 | DP |
| FrameDiff-ICML | ∼7 days | ∼1.9m | 2 | DP |
| FrameDiff-Improved | +7 days | +1.9m | 2 | DP |
| FrameDiff-Retrained | 10 days | 2.2m | 2 | DP |
| FOLDFLOW (BASE, OT and SFM) | ∼2.5 days | 600k | 4 | DDP |



(a) FOLDFLOW       (b) FOLDFLOW-rand       (c) FrameDiff

Figure 12: Ramachandran plot of $\Phi$ and $\Psi$ for (a) FOLDFLOW with informed prior (b) FOLDFLOW-rand with uniformed prior and (c) FrameDiff at 10k steps. With the informed prior, FOLDFLOW is able to better capture the mode on the left.

We chose bovine pancreatic trypsin inhibitor (BPTI) to study in this experiment. The 58-residue protein is the first protein whose dynamics were studied experimentally by nuclear magnetic resonance (NMR) and was the first protein that was simulated by molecular dynamics for 1ms. On timescales ranging from nanoseconds to milliseconds, the dynamics of BPTI involve protein backbone structural changes that, for example, accommodate water molecule exchange and disulfide isomerization (Persson & Halle, 2008). We used the $1-ms$ MD simulation trajectory at a temperature of 300 K (Shaw et al., 2010) to reproduce and interpret the kinetics of folded BPTI. To construct our source distribution, we first generated four folded conformations from each of AlphaFold2, ESMFold, RoseTTAFold, and Unifold (Li et al., 2022). Our source distribution was then added a small amount of noise from the standard Gaussian and $\mathcal{IG}_{\mathrm{SO}(3)}$. We trained FOLDFLOW for one day on 4 A100 GPUs.

Results in Section 5.3 show that FOLDFLOW generates conformations covering all modes of the true conformation distribution. Moreover, we sample different conformations of BPTI from AlphaFold2 and plot them on the Ramachandran and ICA plots, observing while FOLDFLOW can capture all modes of the distribution AlphaFold2 only captures one. Further fig. 11 shows that the KL divergence between the distribution of angles generated by FOLDFLOW is low and uniform, conveying it has learned the distribution of the target. We believe this is an exciting direction meriting larger experiments on more proteins in the future.

Table 5: Quantitative performance on the equilibrium conformation generation task on the BPTI protein. Measures the 2-Wasserstein $\mathcal{W}_2$ in angle space between generated and test samples over all residues ($\mathcal{W}_2$), the most flexible residue ($\mathcal{W}_2$@56), and the Kullback-Leibler divergence also at the most flexible residue (KL @ 56). We also measure the distance of a distribution of AlphaFold2 structures (AlphaFold2) as well as the distance to samples from the trainset.

| | $\mathcal{W}_2(\downarrow)$ | $\mathcal{W}_2$@56($\downarrow$) | KL@56($\downarrow$) |
|---|---|---|---|
| FOLDFLOW | 4.379 | 0.406 | 0.441 |
| FOLDFLOW-Rand | 4.446 | 0.557 | 1.205 |
| FrameDiff | 4.844 | 0.800 | 3.051 |
| RandomPrior | 18.752 | 1.993 | 0.746 |
| AlphaFold2 | 7.298 | 1.917 | 5.724 |
| Trainset | 4.140 | 0.198 | 0.487 |

**Comparison to Uninformed Prior and FrameDiff**. Next we further describe the setup of the comparison experiment in table 3 and table 5 which compares FOLDFLOW with an informed prior, FOLDFLOW with a uniform random prior (FOLDFLOW-Rand) and FrameDiff. We measure the 2-Wasserstein ($\mathcal{W}_2$) distance between generated and test samples either on all 58 residues (denoted $\mathcal{W}_2$) or just on residue 56 (denoted $\mathcal{W}_2@56$), which is the most flexible residue. The $\mathcal{W}_2$ is calculated using the $\mathrm{SO}(2)^{2N}$ distance on $\Phi$ and $\Psi$. We also compare the generated and test samples over the Ramachandran plot of residue 56 with a Kullback-Leibler (KL) divergence (depicted in fig. 12). Here we compute an empirical histogram from samples over a 100 by 100 grid in angle space. We then compute the KL divergence between the smoothed empirical distributions where the minimum of each bin is clipped at $10^{-10}$ for stability.

We train each model for 10k steps on 2 A100 GPUs using either a random prior (uniform over rotations and Gaussian translation) or a mixture of Gaussians / $\mathcal{IG}_{\mathrm{SO}(3)}$ distributions using centers defined by the four folded prior conformations and with standard deviation and $\mathcal{IG}_{\mathrm{SO}(3)}$ concentration 0.5. We do not use inference annealing for this experiment. We generate 250 samples from the model and test against 1000 samples from the test set for computational efficiency reasons.

To contextualize these results, we compare the performance of these models with various baselines such as 250 samples from a random prior (used as the prior in FOLDFLOW-Rand and FrameDiff where each residue is sampled from $\mathcal{N}(0, 10)$), 160 conformations sampled from AlphaFold 2, and 250 samples from the training set (Trainset). Results are averaged over 10 seeds for the random prior and the train set. The trainset represents a well-trained model as the Wasserstein distance is not zero even for empirical distributions drawn from the same distribution. The RandomPrior and the AlphaFold2 represent the random and informed priors respectively. All models are significantly better than these two priors, and FOLDFLOW approaches the performance of samples from the training set.

Figure 12 depicts the Ramachandran plots for residue 56 with scatter plots for FOLDFLOW FOLD-FLOW-Rand and FrameDiff against a kernel density estimate (KDE) of the test set. We see that FOLDFLOW with the informed prior is able to model both modes where FOLDFLOW-Rand and FrameDiff both focus on the mode in the bottom right, centred at $\Phi = \pi/2$.

We have two major findings from this experiment:

- An informed prior helps improve performance both overall and on the most flexible residue as seen by comparing the performance of FOLDFLOW and FOLDFLOW-Rand in table 3.
- FOLDFLOW (with both informed and random priors) improve over FrameDiff on this task.

The equilibrium conformation generation task, studied here, is an example of a setting where an informed prior may be useful. Recent work has explored other applications of starting from an informed prior, such as protein docking (Somnath et al., 2023; Stärk et al., 2023), single-cell (Tong et al., 2023b) and image-to-image translation (Liu et al., 2023b).

## H  FURTHER DISCUSSION OF FOLDFLOW AND RELATED MODELS

**Symmetries as an inductive bias for flow matching**. Leveraging symmetries as an inductive bias in deep learning models (for example by data augmentation or design equivariant models) has been shown to improve data efficiency and lead to better generalization. In the context of flow matching for proteins, the goal is to learn the vector field generating the flow, which maps an invariant source to an invariant target distribution, guaranteeing the existence of an equivariant vector field (Köhler et al., 2020; Bose & Kobyzev, 2021). Therefore, one way to exploit this symmetry would be to parameterize the vector field with an equivariant network, taking as input the 3D coordinates of the protein. Alternatively, since protein backbones can be parametrized by elements of $\mathrm{SE}(3)^N$, we can directly construct the vector field by taking an intrinsic perspective by using charts on the manifold and their coordinate system. In this case, as the vector field lies on the tangent space of $\mathrm{SE}(3)$ it is equivariant by construction.

**Comparison between flow matching and diffusion approaches**. While flow matching and diffusion models bear many similarities they also have key differences which we highlight in this appendix.

1. Flow matching based approaches enjoy the property of transporting any source distribution to any target distribution. This is in contrast to diffusion where one typically needs a Gaussian-like source distribution.

2. Flow matching approaches are readily compatible with optimal transport due to the same property of being able to transport and source to any target. Optimal transport which itself has the advantage of providing faster training with a lower variance training objective and reducing the numerical error in inference due to straighter paths. Diffusion models by themselves are not amenable to optimal transport but instead one can do entropic regularized OT. In Euclidean space, this corresponds to a Schrodinger bridge but this is not an optimal transport path is it stochastic.

3. In general, simulating an ODE is much more efficient than simulating an SDE during inference. Conditional flow-matching and OT-conditional flow matching (Lipman et al., 2022; Tong et al., 2023b) both learn ODEs as the learned flow corresponds to a continuous normalizing flow. Diffusion models on the other hand are SDEs and while being more robust to noise in higher dimensions require more challenging inference.

**Comparison to FrameDiff**. While our model uses a similar setup to FrameDiff, we introduce a number of improvements that help to stabilize training and improve performance. Indeed our additions lead to improvements on all metrics over the FrameDiff-Improved model released on GitHub which substantially improves on the designability over FrameDiff-ICML. We first recap the improvements made in FrameDiff-Improved over FrameDiff-ICML as detected in the code:

1. A bug in the score calculation for rotations means that there is a stop gradient in the rotation score calculation and FrameDiff-ICML is not trained to match the rotation score, which makes its performance quite impressive given this limitation. This bug is fixed in the FrameDiff-Improved model which uses a different score calculation.

2. The dataloader was switched from sampling uniform over proteins in the dataset, to uniform over clusters, then uniform within clusters. As we explore in Appendix I.5, this changes the distribution of proteins but overall increases diversity as there are many similar proteins in a small number of clusters Figure 13c.

3. The rotation loss was changed to use a separate axis and angle component to reduce variance in the loss.

While these items improve the performance of FrameDiff, especially in terms of designability, there are still a few potential areas for improvement.

One area we focus on is the costly loss function of FrameDiff which relies on calculating the pdf of $\mathcal{IG}_{\mathrm{SO}(3)}$ for sampling and computing the score. In the setting used Frame-Diff, the infinite-sum formulation of the density from eq. (26) had to be used, leading to an expensive score loss.

We also noticed that FrameDiff does not exactly follow theory in that the model is not exactly translation invariant: As mentioned in section 3 in order to obtain $\mathrm{SE}(3)$ invariant distributions, the center of mass has to be removed. However, in the FrameDiff code, this was only done at inference and not during training. It is unclear to what extent this impacts the performance, as the model remains translation invariant in expectation and during inference.

## I   IMPLEMENTATION DETAILS AND EXPERIMENTAL SETUP

### I.1   TRAINING AND INFERENCE

To describe the precise algorithm for training FOLDFLOW models over distributions in $\mathrm{SE}(3)^N$. Our starting distribution in $\mathrm{SE}(3)^N$ is $r_1 \sim \mathcal{U}_{\mathrm{SO}(3)}$ i.e. uniform over rotations and $s_1 \sim \mathcal{N}(0, I)$. After centering (i.e. subtracting the mean) this distribution will be uniform over rotations and with translations distributed according to the centered normal $(r_1, s_1^c) \in \mathrm{SE}(3)_0^N$, with $s_1^c \sim \mathcal{N}^c(0, 1)$. In algorithm 1, we also slightly abuse the notation and denote the output of the rotation part of $v_\theta$ as $v_\theta(t, r_t)$ and similarly the translation part of $v_\theta$ as $v_\theta(t, s_t)$. We do not include separate algorithms for FOLDFLOW-BASE and FOLDFLOW-OT as they are simple modifications to FOLDFLOW-SFM. If we set $\gamma_r(t) = 0$ and $\gamma_s(t) = 0$, then we recover the FOLDFLOW-OT algorithm. If in addition we remove the resampling in lines 4 and 5 then we recover the FOLDFLOW-BASE algorithm.

---

**Algorithm 1** FOLDFLOW-SFM training on $\mathrm{SE}(3)^N$

---

1: **Input:** Source and target $\rho_1(x_1), \rho_0(x_0)$, flow network $v_\theta$, and diffusion scalings $\gamma_r(t), \gamma_s(t)$.
2: **while** Training **do**
3: $\quad\quad t, x_0, x_1 \sim \mathcal{U}(0,1), \rho_0, \rho_1$
4: $\quad\quad \bar{\pi} \leftarrow \mathrm{OT}(x_0, x_1)$ $\qquad\qquad\qquad\qquad$ ▷ *OT resampling step to obtain* FOLDFLOW-OT
5: $\quad\quad (r_0, s_0), (r_1, s_1) \sim \bar{\pi}$
6: $\quad\quad s_0^c, s_1^c \leftarrow s_0 - \frac{1}{N}\sum_i s_0^i, \quad s_1 - \frac{1}{N}\sum_i s_1^i$ $\quad$ ▷ *mean subtract:* $(s_0^c, r_0), (s_1^c, r_1) \in \mathrm{SE}(3)_0^N$
7: $\quad\quad r_t \leftarrow \exp_{r_0}(t\log_{r_0}(r_1))$ $\qquad\qquad\qquad$ ▷ *geodesic interpolant from eq. (2)*
8: $\quad\quad s_t \leftarrow ts_1^c + (1-t)s_0^c$ $\qquad\qquad\qquad\qquad$ ▷ *interpolant (Euclidean)*
9: $\quad\quad \tilde{r}_t \sim \mathcal{IG}_{\mathrm{SO}(3)}(r_t, \gamma_r^2(t)t(1-t))$ $\qquad$ ▷ *simulation-free approximation from eq. (9)*
10: $\quad\quad \tilde{s}_t \sim \mathcal{N}(s_t, \gamma_s^2(t)t(1-t))$
11: $\quad\quad \mathcal{L}_{\mathrm{FOLDFLOW}} \leftarrow \left\|v_\theta(t, \tilde{r}_t) - \frac{\log_{\tilde{r}_t}(r_0)}{t}\right\|_{\mathrm{SO}(3)}^2 + \left\|v_\theta(t, \tilde{s}_t) - \frac{\tilde{s}_t - s_0^c}{t}\right\|^2$
12: $\quad\quad \theta \leftarrow \mathrm{Update}(\theta, \nabla_\theta \mathcal{L}_{\mathrm{FOLDFLOW}})$
13: **return** $v_\theta$

---

## I.2 SDE TRAINING AND INFERENCE

In this section, we outline our training and inference algorithms for the $\mathrm{SO}(3)$ component of FOLDFLOW-SFM The training algorithm is detailed in algorithm 2 while the inference algorithm is provided in algorithm 3.

---

**Algorithm 2** FOLDFLOW-SFM training on $\mathrm{SO}(3)$

---

1: **Input:** Source and target $\rho_1, \rho_0$, diffusion schedule $\gamma(\cdot)$, flow network $v_\theta$
2: **while** Training **do**
3: $\quad\quad t, x_0, x_1 \sim \mathcal{U}(0,1), \rho_0, \rho_1$
4: $\quad\quad \bar{\pi} \leftarrow \mathrm{OT}(x_0, x_1)$
5: $\quad\quad r_0, r_1 \sim \bar{\pi}$
6: $\quad\quad r_t \leftarrow \exp_{r_0}(t\log_{r_0}(r_1))$ $\qquad\qquad\qquad$ ▷ *geodesic interpolant from eq. (2)*
7: $\quad\quad \tilde{r}_t \sim \mathcal{IG}_{\mathrm{SO}(3)}(r_t, \gamma^2(t)t(1-t))$ $\qquad$ ▷ *simulation-free approximation from eq. (9)*
8: $\quad\quad u_t(\tilde{r}_t|r_0, r_1) \leftarrow \frac{\log_{\tilde{r}_t}(r_0)}{t}$
9: $\quad\quad \mathcal{L}_{\mathrm{FOLDFLOW\text{-}SFM}} \leftarrow ||v_\theta(t, r_t) - u_t(\tilde{r}_t|r_0, r_1)||_{\mathrm{SO}(3)}^2$
10: $\quad\quad \theta \leftarrow \mathrm{Update}(\theta, \nabla_\theta \mathcal{L}_{\mathrm{FOLDFLOW\text{-}SFM}})$
11: **return** $v_\theta$

---

**Algorithm 3** FoldFlow-SFM Inference

---

1: **Input:** Source distribution $\rho_1$, flow network $v_\theta$, diffusion schedule $\gamma(\cdot)$, inference annealing $i(\cdot)$, noise scale, $\zeta$, integration step size $\Delta t$.
2: Sample $r_1 \sim \rho_1$
3: **for** $s$ in $[0, 1/\Delta t)$ **do**
4: $\quad\quad t \leftarrow 1 - s\Delta t$
5: $\quad\quad$ Sample $z \sim \mathcal{N}(0, 1)$
6: $\quad\quad dB_t \leftarrow \zeta\gamma_t \cdot \sqrt{dt} \cdot z$
7: $\quad\quad d\hat{B}_t \leftarrow \mathrm{hat}(dB_t)$ $\qquad\qquad\qquad\qquad$ ▷ *map rotation vector to* $\mathfrak{so}(3)$
8: $\quad\quad u_t \leftarrow r_t^\top v_\theta(t, r_t)$ $\qquad\qquad\qquad$ ▷ *parallel-transport the vector field to* $\mathfrak{so}(3)$
9: $\quad\quad r_{t+\Delta t} \leftarrow r_t\exp(u_t i_t dt + d\hat{B}_t)$
10: **return** $r_0$

---

## I.3 VECTOR FIELD PARAMETRIZATION

Similar to the toy experiment, for the protein modelling case, the architecture is constructed such that the output vector lies on the tangent space.

### I.3.1 PROTEIN MODEL PARAMETERIZATION

For proteins, we use the FrameDiff architecture (Yim et al., 2023b) over $\mathrm{SE}(3)_0^N$, which is based on the structure module of AlphaFold2 (Jumper et al., 2021) following the initial work on diffusion models with AF2-like architectures (Anand & Achim, 2022). As described in the main text, this architecture $w_\theta$ outputs a predicted $\hat{x}_0$, which we can then deterministically transform into a vector located at the tangent space of $x_t$. This transformation can be split into $2N$ components, the $N\,\mathbb{R}^3$ components, and the $N$, $\mathrm{SO}(3)$ components. For the $\mathrm{SO}(3)$ components we calculate

$$v_\theta(t, r_t) = \frac{\log_{r_t} \hat{r}_0}{t}, \tag{44}$$

and for the $\mathbb{R}^3$ components we calculate after centering,

$$v_\theta(t, s_t) = \left(\frac{s_t - \hat{s}_0}{t}\right) - \frac{1}{N}\sum_i^N (\frac{s_t - \hat{s}_0}{t})_i, \tag{45}$$

where $(\hat{r}_0, \hat{s}_0) = \hat{x}_0 = w_\theta(t, x_t)$. For $\mathbb{R}^3$, $v_\theta(t, s_t)$ is clearly on the tangent space as $\mathbb{R}^3$ is isomorphic to its own tangent space. This is because Euclidean space is a flat space. For $\mathrm{SO}(3)$, $v_\theta(t, r_t)$ is also on the tangent space of $r_t$ due to the definition of the $\log$ map. Since all components of the product space $\mathrm{SE}(3)_0^N$ are on the tangent space, $v_\theta(t, x_t)$ is on the tangent space of $\mathrm{SE}(3)_0^N$.

### I.4 PROTEIN TASK HYPERPARAMETERS

FOLDFLOW is implemented in Pytorch, and uses the invariant point attention (IPA) implementations from OpenFold (Ahdritz et al., 2022) in the backbone. We use the Adam optimizer with constant learning rate $10^{-4}$, $\beta_1 = 0.9$, $\beta_2 = 0.99$. The batch size depends on the length of the protein to maintain roughly constant memory usage. In practice, we set the effective batch size to

$$\mathrm{eff\_bs} = \max(\mathrm{round}(\#GPUs \times 500,000/N^2), 1) \tag{46}$$

for each step. We set $\lambda_{aux} = 0.25$ and weight the rotation loss with coefficient $0.5$ as compared to the translation loss which has weight $1.0$.

We also used a trick from FrameDiff-Improved to stabilize the rotation loss. Instead of the $L^2$ loss on the rotation vector, we separate the loss into two components: one on the axis and one on the angle for the rotation vector. This seemed to reduce variance and numerical instability in the training.

### I.5 DATA AND DATA SAMPLING

We use a subset of PDB filtered with the same criteria as FrameDiff, specifically, we filter for monomers of length between 60 and 512 (inclusive) with resolution $< 5\text{Å}$ downloaded from PDB (Berman et al., 2000) on July 20, 2023. After filtering out any proteins with $> 50\%$ loops we are left with 22248 proteins. To support diversity, we sample uniformly over clusters with similarity of 30% as suggested in FrameDiff-Improved model[2]. Our model functions most efficiently with batches of proteins of the same length, so we each batch contains proteins of a single length. There are 4268 clusters in our dataset.

To assess the effects of our sampling methods on protein diversity and length distribution, we present three plots. fig. 13a illustrates the variability and range of protein lengths in the dataset, giving an overview of available lengths for sampling. fig. 13b shows the batch fraction per length, highlighting alterations in sequence length distribution during training due to uniform cluster sampling. Fig. 13c, which uses a log scale on both axes, unveils the variation in cluster sizes and the skewness in protein distribution across clusters. Uniform cluster sampling enhances batch diversity, aiding model generalization over various protein sequences and structures. However, as observed in fig. 13b, it slightly modifies the sequence length distribution during training. fig. 13c reveals an unevenness in protein distribution across clusters, with two bins containing approximately 14% of proteins.

---

[2]https://cdn.rcsb.org/resources/sequence/clusters/clusters-by-entity-30.txt

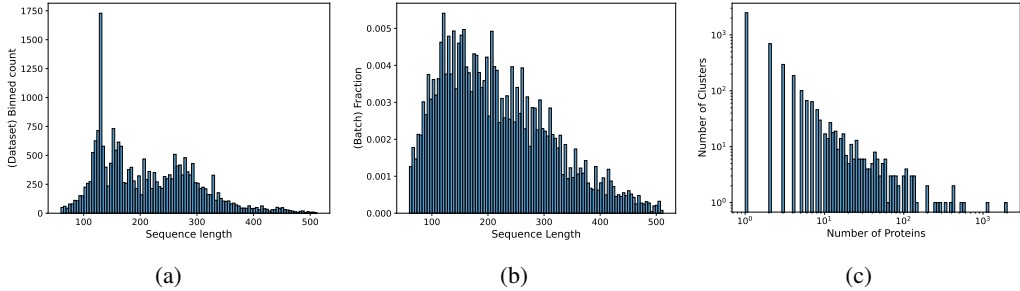

Figure 13: (a) Distribution of protein lengths in our dataset. (b) Distribution of protein lengths in a batch (when sampling uniformly by cluster) (c) Distribution of number of proteins per cluster.

### I.6 PROTEIN METRICS

**The TM-score**. The template modeling score (TM-score) measures the similarity between two protein structures. The TM score can be expressed for two protein backbones $x_0, x_1 \in \mathrm{SE}(3)^N$ as

$$\text{TM-score}(x_0, x_1) = \max \left[ \frac{1}{N_{\text{target}}} \sum_{i}^{N_{\text{common}}} \frac{1}{1 + \left( \frac{d_i}{d_0(N_{\text{target}})} \right)^2} \right] \tag{47}$$

where $N_{\text{target}}$ is the length of the target sequence, $N_{\text{common}}$ is the length of the common sequence after 3D structural alignment, $d_i$ is the distance (post alignment) of the $i^{th}$ residues in $x_0$, and $x_1$, and $d_0(N) = 1.24(N - 15)^{1/3} - 1.8$ is a scaling factor to normalize across protein lengths. The TM-score ranges between $(0, 1]$ with a TM-score of 1 indicating perfectly aligned structure. In general a TM-score $> 0.5$ are considered roughly similar folds, with TM-score $< 0.2$ corresponding to randomly chosen unrelated proteins.

**The RMSD metric**. The root-mean-square deviation (RMSD) is a simple metric over paired residues expressed as

$$\text{RMSD}(x_0, x_1) = \sqrt{\sum_{i=1}^{L} \frac{d_i^2}{L}} \tag{48}$$

where $d_i$ is again the distance between the $i^{th}$ residues heavy atoms $[\mathrm{N}, \mathrm{C}_\alpha, \mathrm{C}, \mathrm{O}]$. The RMSD score is length dependent unlike the TM-score, but has been shown to be a more stringent filtering step then TM-score $> 0.5$ for designability (Watson et al., 2023). In general, as compared to TM-score the RMSD metric is more sensitive local errors and less sensitive to global misalignments.

**Designability**. A generated protein structure is considered designable if there exists an amino acid sequence which refolds to that structure. We first generate 50 proteins at lengths {100, 150, 200, 250, 300 }, then apply ProteinMPNN with sampling_temp = 0.1 8 times to generate 8 sequences for every generated structure. Finally we apply default ESMFold and aligned RMSD of the $C_\alpha$ backbone atoms to calculate alignment of each ESMFold-refolded structure with the generated structure. We determine a protein designable if at least one of the 8 refolded structures has an scRMSD score $< 2.0$. While a threshold of $< 2.0$ for designability is standard, this threshold may be unreasonably strict for longer backbones. However, it is unclear how this threshold should decay with increasing sequence length.

Finally, we note the imperfection of the self-consistency designability metric: when ESMFold does not produce the same structure as FOLDFLOW it does not imply FOLDFLOW's structure is wrong, especially for longer sequences where protein folding models are known to perform worse. Both ProteinMPNN and ESMFold are imperfect, and the failure cases of these models has not been well characterized. While the false positive rate of this metric appears to be low, the false negative of this metric has not been quantified.

**Diversity**. We calculate all pairwise TM-scores for all generated structures that achieve the designability threshold of scRMSD $< 2$ for each length of protein. We then compute the mean over all of these pairwise TM-scores as our diversity metric. For this metric, a lower score is better. We choose to compare diversity on designable proteins as we do not want the designability score to be

inflated by models which produce poor, proteins that may be very dissimilar to the space of refoldable proteins at that length.

**Novelty**. We calculate novelty using two metrics. The first is the minimum TM-score of designable generated proteins to the training data as described in Appendix I.5. The second metric is motivated by previous research (Lin & AlQuraishi, 2023) and is the fraction of proteins that are both designable (scRMSD < 2 Å) and novel (avg. max TM-score < 0.5). We note that all models are not trained on the same dataset: FOLDFLOW and FrameDiff-Retrained share their dataset andFOLDFLOW and FrameDiff-ICML use very similar training datasets (only differing in about 10% of structures),. However, Genie and RFdiffusion use substantially larger datasets. Genie is trained on the Swissprot database (Jumper et al., 2021; Varadi et al., 2021) and RFdiffusion is at least pretrained on high-confidence AlphaFold2 structures. These larger training sets may cause novelty to be overestimated for these models as there are structures in their training sets that are far from the training set we use to test novelty against.

**Error bounds in Table 2**. We also report the standard error of the novelty and designability metrics in table 2. This is calculated by taking the standard error for each metric per sequence length, and then taking the mean over sequence lengths. We note that as the diversity is calculated as the averaged pairwise distances of designable proteins, each estimate of the mean is correlated resulting in an invalid estimate of the standard error.

# J   EXTENDED ABLATION OF EXPERIMENTS

Table 6: Ablation study of FOLDFLOW features (stochasticity, optimal transport, auxiliary losses and inference annealing) against designability, diversity and novelty metrics.

| Designability($\uparrow$) | Diversity($\downarrow$) | Novelty | | Stochas. | OT | Aux. Loss | Inf. annealing |
|---|---|---|---|---|---|---|---|
| | | max($\downarrow$) | fraction($\uparrow$) | | | | |
| 0.228 | 0.230 | 0.440 | 0.172 | ✗ | ✗ | ✗ | ✗ |
| 0.648 | 0.267 | 0.447 | 0.412 | ✗ | ✗ | ✗ | ✓ |
| 0.132 | 0.235 | 0.432 | 0.096 | ✗ | ✗ | ✓ | ✗ |
| 0.657 | 0.264 | 0.452 | 0.432 | ✗ | ✗ | ✓ | ✓ |
| 0.112 | 0.209 | 0.414 | 0.088 | ✗ | ✓ | ✗ | ✗ |
| 0.592 | 0.247 | 0.419 | 0.424 | ✗ | ✓ | ✗ | ✓ |
| 0.152 | 0.190 | 0.443 | 0.108 | ✗ | ✓ | ✓ | ✗ |
| 0.820 | 0.247 | 0.460 | 0.484 | ✗ | ✓ | ✓ | ✓ |
| 0.128 | 0.198 | 0.394 | 0.100 | ✓ | ✗ | ✗ | ✗ |
| 0.580 | 0.253 | 0.439 | 0.416 | ✓ | ✗ | ✗ | ✓ |
| 0.164 | 0.196 | 0.427 | 0.120 | ✓ | ✗ | ✓ | ✗ |
| 0.684 | 0.253 | 0.412 | 0.500 | ✓ | ✗ | ✓ | ✓ |
| 0.188 | 0.215 | 0.449 | 0.136 | ✓ | ✓ | ✗ | ✗ |
| 0.632 | 0.257 | 0.433 | 0.432 | ✓ | ✓ | ✗ | ✓ |
| 0.268 | 0.210 | 0.446 | 0.188 | ✓ | ✓ | ✓ | ✗ |
| 0.716 | 0.251 | 0.411 | 0.544 | ✓ | ✓ | ✓ | ✓ |

We perform a complete ablation experiment for the four features of the FOLDFLOW models: stochasticity, optimal transport, auxiliary losses in training and inference annealing. For each of the experiments, we evaluate the performance of the model on the designability, diversity and novelty metrics. These results can be seen in table 6. Overall, we observe the following trends:

- Stochasticity improves robustness and the ability of the model to generate novel proteins in 7/8 settings, as observed in the fraction of novel and designable proteins (Novelty-fraction).

- Optimal transport improves the designability of the model by reducing the variance in the training objective in 7/8 settings.

- The auxiliary losses improve the designability of the models in 7/8 settings.

- Inference annealing improves the performance of all FOLDFLOW models in all metrics.

