# OpenReview forum: "SE(3)-Stochastic Flow Matching for Protein Backbone Generation"
_ICLR.cc/2024/Conference — ICLR 2024 spotlight_

### Official Review · Reviewer_Nm3A · 2023-10-29

**Soundness:** 4 excellent
**Presentation:** 4 excellent
**Contribution:** 3 good
**Rating:** 8
**Confidence:** 4

**Summary:**

This paper presents FoldFlow models, extensions of Riemannian flow matching objective for distributions over $SE(3)^N$ with applications to protein backbone generation. Specifically, FoldFlow-Base leverages Riemannian FM defined on $SO(3)$. FoldFlow-OT further enhances the approach by Riemannian optimal transport. FoldFlow-SFM introduces stochasticity into the flow model. The models have been comprehensively benchmarked on protein backbone generation tasks and achieved strong performance compared with the diffusion-based baselines.

**Strengths:**

1. The method is very well motivated as an application of flow matching on data with the form of $SE(3)^N$, e.g., protein backbones.

2. The presentation is very clear and the paper is well prepared.

3. The experimental evaluations are interesting and the results are promising.

**Weaknesses:**

1. It would be better if some detailed results on the generation process are presented (see Q1).

2. Ablation studies could be improved to provide a full picture on the proposed techniques/tricks (see Q2).

**Questions:**

1. Is it possible to analyze how the sampling steps affect the quality of the generated samples? Presenting such results would give the readers better insight how the generation process works for this flow-matching based model, especially compared with diffusions (e.g., FrameDiff).

2. For the ablation study, it would be great to have the ablate the four parts (Stochas., OT, Aux loss, inf. annealing) on top of the full version, besides the incremental approach adopted in this paper. That is, it would be interesting to see how the performance would be if we remove  one of the four parts respectively from the full version with all four techniques.

---

> ### Author Response · Authors · 2023-11-19
> **Response**
>
> We are grateful to the reviewer for their time and encouraging comments regarding our manuscript. In particular, we are heartened to hear the reviewer found our work “very well motivated” with a “very clear” presentation. We also value that the reviewer felt that the experimental evaluation is “interesting” and contains “promising” results. We next address the main questions raised by the reviewer.
>
> ## Insight into sampling
>
> This is a great question. We first note that FoldFlow-Base and FoldFlow-OT learn deterministic flows which are ODEs while FoldFlow-SFM and FrameDiff are stochastic and thus correspond to SDEs. However, the SDEs learned during training differ between FoldFlow-SFM and FrameDiff. In a nutshell, FrameDiff centers the diffusion process at the origin and adds time-scaled Brownian noise while FoldFlow-SFM constructs a path between two endpoints where the mean of the IGSO(3) distribution is the McCann (optimal transport) interpolant. Thus, during (sampling) inference, the learned vector field between FoldFlow-SFM and FrameDiff is different and is one of the reasons for the differences in their performance.
>
> Furthermore, when visually inspecting the frames as we move in time during inference we noticed that for large parts of the trajectory, the generated protein lacked a coherent structure. Furthermore, noticeable protein structures—e.g. Helices, sheets—appear near the end of the trajectory. Such visual inspection motivated us to consider our inference annealing trick of $i(t) = ct$ which has the intended effect of speeding up the inference path at the beginning—i.e. when we are starting from noise—and slowing (because of the $t$ factor) near the end of inference. The impact of slowing down allows the model to “dwell” longer near the exact data distribution which helps create higher fidelity samples. We also note that inference annealing helps control the norm on the rotation flow such that it does not explode near the end of inference.
>
>
> We thank the reviewer for their interesting idea of analyzing the performance of the models at different sampling steps. We are currently running experiments to obtain quantitative results on the performance and will update the manuscript with the results as soon as they become available.
>
>  ## Ablation study
>
> We appreciate the reviewer's comment regarding the value of including ablation studies with all the combinations of the 4 features of the model (Stochasticity, OT, Aux. loss, Inf. Annealing). We politely refer the reviewer to our global response regarding the additional ablation experiments we are now including. In summary, we observe that, as expected, OT improves designability, Stochasticity improves robustness and generalization, and inference annealing improves the overall performance of all of the models. The results of our ablation experiments are included in Table 5 in the appendix. We are also now re-training our models without the auxiliary loss and will include those results in the final manuscript as well.
>
> We thank the reviewer again for their suggestions that helped us strengthen our ablation study. We hope that our rebuttal responses fully addressed all the important questions raised by the reviewer and we kindly ask the reviewer to potentially upgrade their score if the reviewer is satisfied with our responses. We are also more than happy to answer any further questions that arise.

---

> > ### Comment · Reviewer_Nm3A · 2023-11-20
> > **Response**
> >
> > Thank the authors for the response. I remain my positive score for the paper, while strongly recommending the authors include the suggested additional ablation studies as promised.

---

> > > ### Author Response · Authors · 2023-11-21
> > > **Final Ablation Results**
> > >
> > > We thank the reviewer for their helpful comments and positive feedback.
> > > We have now included all ablation studies, including those with auxiliary losses, in the updated Table 6 of Appendix J. Overall, we observe the following trends:
> > >
> > >  - Stochasticity improves robustness and the ability of the model to generate novel proteins in $7/8$ settings, as observed in the
> > >     fraction of novel and designable proteins (Novelty-fraction).
> > >  - Optimal transport improves the designability of the model by reducing the variance in the training objective in $7/8$ settings.
> > >  - The auxiliary losses improve the designability of the models in $7/8$ settings.
> > >  - Inference annealing improves the performance of all $8/8$ FoldFlow models in all metrics.
> > >
> > > We believe that these results significantly strengthen our empirical claims and thank the reviewer for their suggestion.

---

> > > > ### Comment · Reviewer_Nm3A · 2023-11-22
> > > > **Response**
> > > >
> > > > I thank the authors for providing the complete results. Good work.

---

### Official Review · Reviewer_zh9j · 2023-10-30

**Soundness:** 4 excellent
**Presentation:** 3 good
**Contribution:** 4 excellent
**Rating:** 8
**Confidence:** 3

**Summary:**

The authors extend previous work on conditional flow matching on Riemannian manifolds and apply it to generate protein backbones. Their theoretical results lead to fast and stable training, and they empirically generate more diverse and designable structures than non-pretrained denoising diffusion probabilistic models of protein backbone.

In general, I think this is an interesting and strong paper. Most of my concerns (detailed below) are around specifics of the evaluation, and I believe that they are addressable

**Strengths:**

- \[originality\] This is in the first cohort of work applying conditional flow matching to generate protein backbones. The paper makes a number of theoretical contributions, such as proving the existence of a Monge map on $SE(3)_0^N$ and a closed-form expression of the target conditional vector field for SO(3).
- \[significance\]  Conditional flow matching is a family of generative models that relaxes many of the constraints of normalizing flows, and extending them to SE(3) enables many applications in biology, chemistry, and materials. The specific application studied here is of great importance and interest.
- \[quality\] The theoretical results seem sound, although I did not carefully check the proofs. Empirically, FoldFlow does generate more designable, diverse, and novel structures than previous comparable methods. The comparisons between FoldFlow-Base, OT, and SFM are thorough and sound.
- \[clarity\] The paper is generally well-written, the contributions are clear, and the empirical results are easy to follow. Despite an obvious need to compress a lot of material into the page limit, the paper is well-cited and well-situated in the current literature.

**Weaknesses:**

The main empirical claim in this paper is that using flow matching instead of diffusion results in better generated protein structure backbones. Therefore, the major weaknesses of the paper are in explaining why we see this result and in showing, with as many other parameters held constant as possible, that the flow matching objective is responsible for improvements in generation quality. In addition, the section on conformation generation lacks the proper ablations to support the claim that flow matching is helpful there, as well as a justification for why this is a useful task.

I will now try to detail these weaknesses in clarity and quality and suggest ways to address them where applicable. Within each section, I will move from more important to less important.

### Clarity

- While the theoretical contribution lies in enabling stochastic flow matching for protein backbone generation, there is no theoretical or qualitative discussion of why flow matching should result in more efficient training and higher-quality generations.
- In general, the authors have the unenviable task of explaining the background for both protein generation and Riemannian flow matching within the page limit. I think people from outside the fields will have a lot of trouble following the more technical parts of the exposition, but I'm not sure it's possible to do much better given the depth of material and the page limit.
- Is generating from the equilibrium distribution over conformations after training on data from molecular dynamics useful? I can see how this would be useful if it generalized to new structures, but it seems that if you've already done the MD for that particular model, using another model to sample from the distribution is superfluous.
- There is some information relegated to the supplement that should be in the main text. For example, that the authors use the same neural architecture as FrameDiff but a slightly different training set is very important for interpreting the empirical results.
- The two paragraphs beginning "Our approach" and the final list of main contributions at the end of the introduction are fairly redundant, with several sentences of repeated content. Given how much theory, experiment, and background the authors need to cover, reducing redundancy here could help clarify other parts of the paper.
- In section G.1.1, I think "Here we find that FOLDFLOW is over 2x faster than FoldFlow-Improved per step" should be "Here we find that FOLDFLOW is over 2x faster than FrameDiff-Improved per step."


### Quality

- The claim that FoldFlow outperforms FrameDiff in particular and diffusion in general would be stronger if the training set and other train compute were matched between FrameDiff-improved and FoldFlow. As it is, they are very similar, but the slight discrepancy in train sets and hyperparameters such as batching weaken the comparison.
- The section on equilibrium conformation generation doesn't make a strong case that FoldFlow is better than (1) diffusion from a simple prior trained on the same data or (2) flow matching from a simple prior trained on the same data.
- In Table 2, the numbers for Diversity and novelty should include uncertainties. It would also be nice to have scRMSD in this table.
- Ablations of FoldFlow-OT and -SFM without the Aux loss or inference annealing would strengthen the case that the auxiliary loss and inference annealing are universally helpful.
- FoldFlow-Base and FoldFlow-SFM do poorly on designability for longer proteins. This may be because the deduplicated dataset does not have very many longer proteins.

**Questions:**

- Are there applications for the arbitrary starting distribution other than trying to sample from the conformational ensemble via generation?
- Is there a reason to prefer scRMSD instead of scTM, as was seen in some earlier work?
- Is DP vs DDP something intrinsic to the models, or could, for example, FrameDiff be trained with DDP?

---

> ### Author Response · Authors · 2023-11-19
> **Response Part 1/3**
>
> We thank the reviewer for their detailed review and constructive feedback. We appreciate that the reviewer has found our work “original” and the protein design application that we tackle “significant” and of great importance. Below, we provide a detailed response to the questions raised by the reviewer.
>
> ## Points raised under “Clarity”:
> **Flow matching vs. Diffusion**
>
> We thank the reviewer for their interesting question regarding the motivation behind using flow matching. We have extended our Appendix H to further discuss the theoretical and practical advantages of flow matching over diffusion and have also further motivated the OT and stochastic approaches in sections 3.2 and 3.3. We provide a summary of these points below:
> - Flow matching-based approaches enjoy the property of transporting any source distribution to any target distribution. This is in contrast to diffusion where one typically needs a Gaussian-like source distribution.
> - Flow matching approaches are readily compatible with optimal transport due to the same property of being able to transport and source to any target. Optimal transport which itself has the advantage of providing faster training with a lower variance training objective and reducing the numerical error in inference due to straighter paths. Diffusion models by themselves are not amenable to optimal transport but instead one can do entropic regularized OT. In Euclidean space, this corresponds to a Schrodinger bridge but this is not an optimal transport path as it is stochastic. However, we note that SDEs can leverage OT formulations in the form of entropy-regularization.
> - In general, simulating an ODE is much more efficient than simulating an SDE during inference. Conditional flow-matching and OT-conditional flow matching [3, 4] both learn ODEs as the learned flow corresponds to a continuous normalizing flow. Diffusion models on the other hand are SDEs and while being more robust to noise in higher dimensions require more challenging inference.
> In addition to the points above, an advantage of all of our models over the diffusion approach is that it does not rely on the costly calculation of the pdf of the IGSO(3) distribution, which FrameDiff requires for sampling and computing the score. FoldFlow simply needs to sample from the IGSO(3) which is fast to do.
>
> **Background and Theory**
>
> We understand the reviewer’s point regarding the amount of complex background and theory that we had to cover in this paper. We tried our best to make the information as accessible as possible, especially by including more background material in the appendix and adding references to seminal textbooks on Lie groups and Riemannian geometry. We would be more than happy to include additional details on any of the sections that the reviewer found lacking. Finally, we thank the reviewer for pointing out the potential redundancy between our “main approach” section and contribution list. We have combined and shortened these sections in the updated PDF.
>
> **Utility of Generative Models for MD Simulations**
>
> We acknowledge the reviewer's comment regarding the utility of a generative model for simulating trajectories for proteins. Indeed, as the reviewer correctly points out ideally, the generative model should generalize to MD trajectories of new proteins, which is an active part of our research. We further note that we compare our inference against MD frames not in the training set as well. Nevertheless, considering how expensive MD simulations are, the ability to cheaply simulate such trajectories with a generative model trained on relatively few frames is still of great interest. There has been a substantial amount of work in the Botlzmann generator community on this particular problem (e.g. see [1, 6]), and reducing the cost has practical applications for computational biologists. This also dovetails with the larger goal of AI4Science which seeks to amortize the cost of running expensive simulations with AI-accelerated approximations to achieve faster inference.
>
> **Architecture and training set compared to FrameDiff**
>
> We thank the reviewer for their comment. To enable the fairest comparison possible, we have now retrained FrameDiff using this new training set (which we call FrameDiff-Retrained) and have included the updated results in the main text. In section 4 under Architecture, we mention that we follow Yim et al. (2023b) in our architecture. We have further highlighted this by explicitly mentioning FrameDiff in the text.
>
> **Typo** We thank the reviewer for pointing out the typo in G.1.1. It has been fixed.
>
> Part 1/3

---

> > ### Author Response · Authors · 2023-11-19
> > **Response Part 2/3**
> >
> > ## Points raised under “Quality”:
> > **Comparison with FrameDiff**
> >
> > We thank the reviewer for their comment regarding the discrepancy between FoldFlow and FrameDiff training sets. As mentioned above, we have included new results on FrameDiff trained with this new dataset (called FrameDiff-retrained). Additionally, we inherit as many hyperparameters as possible from FrameDiff as we have used their code and architecture as our base code. Given that the experimental setup of FrameDiff-retrained and the FoldFlow models resemble one another as much as possible, we believe that we have provided the fairest comparison possible.
> >
> > **Improvements to Table 2**
> >
> > We appreciate the reviewer’s suggestions for improving Table 2. This table now provides a more complete picture of our results, as we have included the scRMSD, an additional metric for measuring novelty as well as standard errors for novelty and designability. Regarding the scRMSD, we can see that they have a similar trend to the designability metric that we originally reported. However, we would like to note that the average scRMSD is mostly affected by a few long proteins in the dataset which is why we originally only included the one that measures the fraction of proteins with scRMSD < 2. We kindly refer the reviewer to our global response for a discussion of the additional novelty metric.
> >
> > **Ablations**
> >
> > We thank the reviewer for their suggestion to include ablations of FoldFlow-OT and -SFM without inference annealing and/or the auxiliary loss. We politely refer the reviewer to our global response regarding the additional ablation experiments we are now including. In summary, we observe that as expected, OT improves designability, Stochasticity improves robustness and generalization, and inference annealing improves the overall performance of all of the models. We are also now re-training our models without the auxiliary loss and will include those results in the final manuscript as well. The results of our additional ablation experiments are included in Table 5 in the appendix.
> >
> > **Ablations for Equilibrium Conformation Generation**
> >
> > We thank the reviewer for their suggestion on including additional ablations for FoldFlow in our equilibrium conformation sampling experiment. We have now included 2 new baselines presented in Table 3 of the main paper. These are FoldFlow-Rand and FrameDiff, both of which use a random prior as opposed to FoldFlow which starts from the empirical dataset. We discuss these results in our global response to all reviewers. In summary, our quantitative results using the Wasserstein-2 distance, as observed in the new Table 3, strengthen our original claim as we find FoldFlow to be the best model and, equally importantly, both FoldFlow and FoldFlow-Rand outperform FrameDiff. We hope that these additional results help answer the reviewer's great question and allow the reviewer to more enthusiastically endorse our experimental findings.
> >
> > **Performance on longer proteins**
> >
> > We agree with the reviewer that the inferior performance of FoldFlow models on longer proteins may be due to a lack of data. We believe that trying out larger proteins is an exciting direction for future work.
> >
> > Part 2/3

---

> > > ### Author Response · Authors · 2023-11-19
> > > **Response Part 3/3**
> > >
> > > ## Other Questions:
> > > **Other Applications of Starting from Arbitrary Distributions**
> > >
> > > We thank the reviewer for their question. Other biological applications can also benefit from starting from a different source distribution. One important example is protein docking where one knows the starting point of the trajectory [5]. Thus, in this case, we can set the source distribution to be the separate unbound conformations, with the target being the true bound conformation.
> > >
> > > **scRMSD vs scTM**
> > >
> > > We thank the reviewer for this interesting question. We chose to report the scRMSD instead of scTM since previous research [2] has shown it to be a more stringent metric and more correlated to the experimental performance of protein designs.
> > >
> > > **DDP**
> > >
> > > We thank the reviewer for the question. DDP refers to Distributed-Data-Parallel and can be used in any model, including FrameDiff. However, modifying the original codebase to support this is non-trivial.
> > > We thank the reviewer for their time and effort in reviewing our work and we hope the reviewer would kindly consider a fresh evaluation of our work given the main clarifying points outlined above as well our global response to all reviewers. We are also eager to engage in further discussion if the reviewer has any lingering doubts.
> > >
> > > ## References
> > >
> > > [1] Frank Noé, Simon Olsson, Jonas Köhler, & Hao Wu. (2019). Boltzmann Generators – Sampling Equilibrium States of Many-Body Systems with Deep Learning.
> > >
> > > [2] Watson, J.L., Juergens, D., Bennett, N.R. et al. De novo design of protein structure and function with RFdiffusion. Nature 620, 1089–1100 (2023).
> > >
> > > [3] Alexander Tong, Nikolay Malkin, Guillaume Huguet, Yanlei Zhang, Jarrid Rector-Brooks, Kilian Fatras, Guy Wolf, and Yoshua Bengio. Improving and generalizing flow-based generative models with minibatch optimal transport. arXiv preprint 2302.00482, 2023b.
> > >
> > > [4] Yaron Lipman, Ricky T. Q. Chen, Heli Ben-Hamu, Maximilian Nickel, and Matt Le. Flow matching for generative modeling, October 2022.
> > >
> > > [5] Octavian-Eugen Ganea, Xinyuan Huang, Charlotte Bunne, Yatao Bian, Regina Barzilay, Tommi Jaakkola, & Andreas Krause. (2022). Independent SE(3)-Equivariant Models for End-to-End Rigid Protein Docking
> > >
> > > [6] Midgley, Laurence Illing, et al. "Flow annealed importance sampling bootstrap." arXiv preprint arXiv:2208.01893 (2022).
> > >
> > > Part 3/3

---

> ### Comment · Reviewer_zh9j · 2023-11-20
> **Response to authors**
>
> The additional baselines and comparisons, especially to their version of FrameDiff trained on the same dataset, greatly strengthen the empirical claims in the paper. It's very interesting that training on their dataset substantially increases sample quality at the expense of diversity and novelty. Adding scRMSD to Table 2 also highlights the large lead RFdiffusion has here, but I think the paper is better off being upfront here about room for further improvements. I'm also surprised that Genie and FrameDiff use DP instead of DDP, and I commend the authors on using the recommended parallelization framework in PyTorch. The new ablations resolve my concerns about soundness.
>
> I would prefer to see some of the intuition behind using flow matching instead of diffusion in the main paper. However, I understand that space is very limited, and that this is not my paper, and I will leave that decision to the authors. I'm not convinced that the dynamics results are useful as is, but with the new ablations and text, I do believe that they are clearly communicated, sound, and an improvement over previous work.
>
> Overall, I have changed my scores from:
>
> Soundness: 2 fair
> Presentation: 3 good
> Contribution: 4 excellent
> Recommendation: 5
>
> to:
>
> Soundness: 4 fair
> Presentation: 3 good
> Contribution: 4 excellent
> Recommendation: 8

---

> > ### Author Response · Authors · 2023-11-21
> > **Thank you!**
> >
> > We thank the reviewer for engaging in this discussion. We greatly appreciate your comments that enabled us to make this paper better and allowed you to upgrade your score. We also find it interesting that this dataset results in higher sample quality and worse diversity and novelty for FrameDiff. We are curious if this holds more generally and why this is the case, but leave this to future work. We have two updates that may interest the reviewer:
> >
> >  - We have now included all ablation studies, including those with auxiliary losses, in the updated Table 6 of the appendix.
> >  - We have also performed additional analysis to show the significance of the equilibrium conformation generation experiment
> >    which are included in an updated Appendix G.2 in Table 5 and Figure 12. We hope that these results further highlight the
> >    significance of this experiment.
> >
> > We thank the reviewer again for their time and valuable feedback.

---

### Official Review · Reviewer_PWT4 · 2023-10-31

**Soundness:** 4 excellent
**Presentation:** 4 excellent
**Contribution:** 4 excellent
**Rating:** 8
**Confidence:** 4

**Summary:**

Flow matching, and OT flow matching are applied to the problem of learning generative models of protein backbones. Three variants of the method are provided (1) train a CNF via flow matching, (2) train a CNF via conditional flow matching with minibatch optimal transport, and (3) they replace the deterministic flow model with a model that learns to map between the base and target sources via stochastic dynamics. Strong empirical results are shown when training on structures from the PDB:
- The OT version shows significant improvement above the base model
- They improve upon FrameDiff substantially.
- Their model does not rely on a pretrained model, and is substantially cheaper to train than RosettaDiff

Additionally, the flow matching paradigm allows for an arbitrary base distribution to be used - and this is demonstrated in an experiment on conformer generation of the BPTI protein where noised samples from pre-trained models are used as the base distribution.

**Strengths:**

- Well motivated: The presentation of the method is clear, the method (flow matching/OT flow matching) is well suited to the problem of protein generation, and is the first to explore this. The ability to use arbitrary base distributions is well suited to confomer generation where we can choose informed base distributions (e.g. using cheminformatics methods, or with existing models e.g. AlphaFold).
- Strong experimental section: The results (improving upon the FrameDiff model) look really good. The models are trained for substantially less time than the RFDiffusion alternative, and do not rely on pre-training. Good baselines are presented, showing the benefit of the OT-flow over the flow trained with vanilla flow matching.

**Weaknesses:**

- The FOLDFLOW-SFM does not have much motivation provided (besides empirical results e.g.  noting that they have higher novelty).
- The experiment on equilibrium confomer generation utilizing the informed base distribution does not compare against an uninformed base distribution. Thus, it is hard to judge exactly how helpful this is (although intuitively it seems like it would be very helpful!).

**Questions:**

- What is the advantage of the FOLDFLOW-SFM over the other models in theory?
- I find the term “stochastic flow” a bit confusing as typically flow refers to deterministic dynamics - could the authors please clarify?

---

> ### Author Response · Authors · 2023-11-19
> **Response**
>
> We thank the reviewer for their time, detailed review, and positive appraisal of our work. We are glad that the reviewer finds our paper “well-motivated”, and believes that flow matching is “well suited” to tackle protein backbone generation and our method FoldFlow is the “first to explore this.” We also appreciate that the reviewer felt that our paper had a “strong experimental section” with “really good” results improving over FrameDiff and other strong baselines which also highlight the benefit of “OT-flows”. We now address the key clarification points raised by the reviewer.
>
>
>
>
>
>
> ## FoldFlow-SFM
> We value the reviewer's concern regarding the insufficient motivation for FoldFlow-SFM in comparison to FoldFlow-Base and FoldFlow-OT. We would like to point the reviewer to our global response to all reviewers in which we give a detailed motivation on the utility of FoldFlow-SFM, and note that we have also highlighted the motivation behind the stochastic approach in section 3.3 of the paper. In summary, as previous research suggests, we expect stochasticity to help with robustness, particularly in high dimensions. This expectation is validated by our updated inference and new results, which demonstrate FoldFlow-SFM’s superior performance in novelty, compared to the deterministic variants of FoldFlow.
>
> ## Equilibrium conformer generation
> We thank the reviewer for their great suggestion regarding the use of an uninformed (random) prior distribution as a baseline for FoldFlow. We have now included new baselines with FoldFlow-Rand and FrameDiff in Table 3 of the main paper. We kindly point the reviewer to our global response for a detailed discussion of these results but in summary, our quantitative numbers (calculated using 2-Wasserstein distance) convincingly show the importance of starting from an informed prior and the continued superiority of FoldFlow over FrameDiff in this new experimental setting.
>
>
> ## Stochastic Flow Terminology
> We appreciate the reviewer's feedback regarding the usage of the term stochastic flow for our FoldFlow-SFM model. Our naming choice is motivated by the fact that FoldFlow-SFM utilizes the (OT) geodesic path between two samples $r_0 \sim \rho_0$ and $r_1 \sim \rho_1$. This path is the OT Flow between these two points and is also the mean of the IGSO(3) distribution. By sampling a noisy point from this IGSO(3) distribution we get a stochastic path which we dub as the $\textit{stochastic flow}$. Note the flow is still the mean of the distribution and the stochasticity simply refers to the injection of IGSO(3) noise to the mean path (flow). We hope this helps clear up our name choice.
>
>
> We thank the reviewer for their time and effort in reviewing our work, and we hope our efforts to clarify the main points along with the global response allow the reviewer to consider improving their score. We are also more than happy to answer any other questions that arise.

---

> > ### Comment · Reviewer_PWT4 · 2023-11-20
> >
> > Thank you for the clarification and further experiments - overall I think this is a good paper and recommend acceptance.
> >
> > For the equilibrium conformer generation experiments it is not very clear to me how significant the improvement relative to the FoldFlow-rand is. Would it be possible to add more evaluation metrics to compare the different methods? For example, (1) I think plotting the Ramachandran plots for the various models may be more intuitive to interpret than the raw Wasserstein metrics, (2) would it be possible to compare the NLL of a test set with the different models?

---

> > > ### Author Response · Authors · 2023-11-21
> > > **Additional Equilibrium Conformation Generation Results**
> > >
> > > We appreciate the reviewer’s thoughtful comments and continued participation in this discussion. To highlight the significance of our equilibrium confirmation generation sampling experiments, we have—as the reviewer rightly encouraged:
> > >
> > >  - Added Ramachandran plots for FoldFlow, FoldFlow-Rand, and FrameDiff.
> > >  - Added the associated KL divergence for residue 56 to the test set, all evaluated at 10k training steps.
> > >  - Added further context to the 2-Wasserstein numbers by adding evaluations of the distance of the training set, random prior, and AlphaFold2 structures to the test set.
> > >
> > > We observe that the FoldFlow model trained with an informed prior achieves a 2-Wasserstein of $4.379$. This is only slightly worse than that of the 2-Wasserstein of the train set to the test set, which is $4.140$, and significantly better than the other models including FoldFlow-Rand at $4.446$. We hope that this clarifies the significance of starting from an informed prior. These results are in an updated Appendix G.2 in Table 5 and Figure 12.
> > >
> > > Analyzing the Ramachandran plot results in Figure 12 we find that FoldFlow, in comparison to FoldFlow-Rand and FrameDiff, is already starting to capture both modes of the distribution, whereas the models with an uninformed prior only focus on one of the modes. Measuring this quantitatively, we find the following KL divergences for residue $56$: $0.441 < 1.205 < 3.051$ for FoldFlow, FoldFlow-Rand, and FrameDiff, respectively.
> > >
> > > We thank the reviewer again for allowing us to improve our paper and we hope that these results, in conjunction with the totality of all the rebuttal updates, allow the reviewer to more enthusiastically endorse our paper and potentially upgrade their score.

---

> > > > ### Comment · Reviewer_PWT4 · 2023-11-21
> > > >
> > > > Thank you for adding these results - they are informative and the FoldFlow does indeed look better (although FoldFlow-Rand does seem to also be capturing both modes, albeit less well). I am going to stick with my score of 8 - although I do strongly endorse this paper and think it has made a very significant contribution.

---

### Official Review · Reviewer_4M3T · 2023-11-08

**Soundness:** 2 fair
**Presentation:** 3 good
**Contribution:** 3 good
**Rating:** 8
**Confidence:** 4

**Summary:**

The paper introduces a series of novel generative models called FoldFlow, which are designed to accurately model protein backbones. The models are based on the flow-matching paradigm over rigid motions, allowing for the construction of stable and fast training methods.
The first model introduced is FoldFlow-Base, a simulation-free approach that learns deterministic continuous-time dynamics and matches invariant target distributions. This model is then improved upon by incorporating Riemannian optimal transport, resulting in FoldFlow-OT, which creates more simple and stable flows.
Lastly, the authors design FoldFlow-SFM, which combines Riemannian optimal transport and simulation-free training to learn stochastic continuous-time dynamics.

**Strengths:**

The paper introduces a novel approach called FoldFlow for generative modeling of protein structures. This approach is based on the flow-matching paradigm over rigid motions. The authors claim that incorporation of Riemannian optimal transport and simulation-free training techniques improves the stability and efficiency of the models compared to diffusion-based approaches.
The authors present a series of models within the FoldFlow framework, each with increasing modeling power.

**Weaknesses:**

The work raised a couple of questions.

1. Why should optimal transport be a good prior for protein backbone generation? Section 3.2 on FoldFlow-OT provides little explanation why this should overall improve the quality of the generated samples. Could you elaborate on that? Similarly, it would be great if you could provide an intuition for motivating the introduction of the stochastic version. The results (Table 2) also do not suggest that there is any benefit in introducing three variants.

3. Studying Table 2, the authors confirm that FoldFlow significantly underperforms RFDiffusion. Also, as raised in Question 1, there is no pattern of improvement between each method of FoldFlow. My main concern regards the stability of the results? You seem to report a single value, is this a lucky single shot. I would suggest to align the analysis to previous literature, i.e., as conducted in FrameDiff, and report results on several runs and examples. It would also be important to also extend Figure 4b to include a comparison to RFDiffusion, FrameDiff, and Genie in terms of scRMSD. Similarly, why not repeat the analysis in Figure 5 with the considered baselines?

3. You claim that FoldFlow models are designed to be fast, stable, and efficient training. Why are the runtimes of the baselines in Table 2 missing? Regarding Table 3, what are the time differences between -base, -OT, and -SFM? What does FoldFlow in Table 3 refer to? You claim in Appendix B that you "want to get straighter flows for faster inference and more stable training". Is this something you observe?

4. Using flow matching methods on protein structures is not new. It would be important to cite previous work here and highlight the differences [1].

I very much enjoyed reading the paper and the direction is exciting. The current experimental validation, however, raises several concerns that need to be addressed. This in particular concerns the absence of a pattern in performance of different FoldFlow variants and chosen reported metrics and comparisons.

[1] Somnath, Vignesh Ram, et al. "Aligned Diffusion Schrödinger Bridges." Conference on Uncertainty in Artificial Intelligence (UAI), 2023.

**Questions:**

See above. If my concerns are addressed, I am willing to increase my score.

---

> ### Author Response · Authors · 2023-11-19
> **Response Part 1/2**
>
> We thank the reviewer for their detailed review and nuanced comments. We appreciate that the reviewer felt our FoldFlow was a “novel” approach for modeling protein backbones. We now address the key clarification points raised in the review.
>
> ## Motivations for using Optimal Transport
> The reviewer raises an important question regarding the importance of OT as a prior for protein backbones. There are several theoretical reasons why considering OT paths is interesting in the generative modeling setup. First note, that any coupling that builds paths between two samples $r_0 \sim \rho_0$ and $r_1 \sim \rho_1$ that minimizes the OT cost necessarily constructs a “shorter” path than a random coupling. Globally, this means the average path length of the flow between $\rho_0$ and $\rho_1$ is shorter, and more importantly, none of the paths cross each other. This is a universal property of OT paths and can be applied to any generative modeling problem—including the protein backbone generation problem we consider in this paper.
> In the specific context of flow matching using OT, existing work in Euclidean space has already demonstrated the empirical benefits of OT paths [1, 2]. Moreover, in the same prior work, OT was shown to lead to faster training with a lower variance training objective in Euclidean spaces [1,2]. This is particularly relevant for protein backbones as we model them as objects in $\text{SE(3)}$ which is the semi-direct product of $\mathbb{R}^3$ and $\text{SO(3)}$. Moreover, for proteins having shorter inference directly corresponds to higher quality and more designable generated samples. Consequently, we wish to inherit and extend the benefits of OT for flow matching in Euclidean space to flow-matching on $\text{SE(3)}$ which is the providence for our FoldFlow family of models. We hope this helps address the reviewer's concerns regarding the motivations for using OT and we hope this matches expectations given the strong performance of FoldFlow-OT on the designability of proteins.
> We have further highlighted the benefits of OT in section 3.2 of the paper.
>
> ## Motivations for FoldFlow-SFM
> We appreciate the reviewer's comment regarding the motivation behind the SFM model and how this gets translated into the experiments. We would like to point the reviewer to our global response in which we give a detailed motivation on the utility of FoldFlow-SFM. In summary, as previous research suggests, we expect stochasticity to help with robustness, particularly in high dimensions. This expectation is validated by our updated results, which demonstrate FoldFlow-SFM’s superior performance in both of our novelty metrics, compared to the deterministic variants of FoldFlow. These results firmly establish FoldFlow-SFM as the most novel and designable model which is a key goal in de novo protein design and AI-accelerated drug discovery at large.
> We have also extended the discussion on the advantages of the stochastic approach in section 3.3 of the paper.
>
> Part 1/2

---

> > ### Author Response · Authors · 2023-11-19
> > **Response Part 2/2**
> >
> > ## Performance Metrics
> > We thank the reviewer for their question regarding our analysis and the reported metrics.
> > - Regarding FoldFlow’s performance compared to RFDiffusion, we would like to reiterate that this is expected given that RFDiffusion uses:
> >    - a pre-trained backbone
> >    - a significantly larger model of 60m parameters (vs our 17m parameters)
> >    - a significantly larger and different training set (at least 2 orders of magnitude bigger), ([3] section 1.4)
> >    - larger compute resources (1800 vs. 10 GPU days).
> >
> >     We also note that, similarly, all versions of FrameDiff underperform RFDiffusion. Given these differences in resources, we believe that FrameDiff is the most appropriate baseline for a rigorous and fair comparison.
> >
> > - Regarding reporting a single value vs. an average over multiple runs, we note that in fact, we follow FrameDiff on how our values are reported. FrameDiff’s reported metrics are also a single value as opposed to an average over multiple runs. This is due to the fact that training these models is computationally expensive and obtaining uncertainty quantification while important is difficult with limited computational resources. Therefore we believe that we have provided a fair comparison to the baseline models.
> > - Regarding the importance of extending our figure on the ablation results of inference annealing (fig. 4b) to other baseline models, we politely disagree. While our technique of inference annealing is interesting and could also be applied to other baselines, reproducing those exact experiments on this dataset is beyond the scope of ‌this work.
> > - With regards to our Equilibrium conformation generation experiment we have added 2 new baselines: 1.) FoldFlow-Rand and FrameDiff both of which start from an uninformed (random) prior distribution. These results are included in a new Table 3 in the main paper and the analysis is presented in the global response. In summary, our FoldFlow model which starts from the empirical data convincingly outperforms these new baselines.
> >
> > ## Training Speed and Runtimes
> > - In Table 2, we only include iters / second comparison to models with the same architecture. We have made a note of this in the manuscript. RFDiffusion and Genie baselines both use a different architecture confounding this comparison. We also note that the training code for RFDiffusion is not publicly available rendering this comparison impossible.
> > - In Table 2 we show that the three FoldFlow models have approximately the same training speed, therefore, to avoid redundancy, in Table 4 we have used the term FoldFlow as an umbrella term for all the models. We thank the reviewer for pointing out how this could be confusing and have changed it to be more clear in the updated manuscript.
> >
> > ## Citation
> > - We thank the reviewer for bringing this important reference to our attention and have added a citation to it. We would like to however politely point out that this paper is concerned with protein docking in Euclidean, which is a different problem than the one considered here. Nevertheless, we agree with the reviewer about the relevance of this paper and we have included it in our related work section.
> >
> > We thank the reviewer for their valuable feedback and great questions. We hope that our rebuttal fully addresses all the important salient points raised by the reviewer and we kindly ask the reviewer to potentially upgrade their score if they are satisfied with our responses. We are also more than happy to answer any further questions that arise.
> >
> > [1] Aram-Alexandre Pooladian, Heli Ben-Hamu, Carles Domingo-Enrich, Brandon Amos, Yaron Lipman, & Ricky T. Q. Chen. (2023). Multisample Flow Matching: Straightening Flows with Minibatch Couplings.
> >
> > [2] Alexander Tong, Nikolay Malkin, Kilian Fatras, Lazar Atanackovic, Yanlei Zhang, Guillaume Huguet, Guy Wolf, and Yoshua Bengio. Simulation-free schrödinger bridges via score and flow matching, 2023a.
> >
> > [3] Watson, J.L., Juergens, D., Bennett, N.R. et al. De novo design of protein structure and function with RFdiffusion. Nature 620, 1089–1100 (2023).
> >
> > Part 2/2

---

> > > ### Author Response · Authors · 2023-11-21
> > >
> > > Dear Reviewer,
> > >
> > > Thank you for your time! We hope that we have addressed all of your concerns in our response.
> > >
> > > We have also updated the manuscript with additional analysis for the equilibrium conformation generation (Table 5, Figure 12) as well as a complete ablation experiment (Table 6). We believe that the additional ablation results show a clear pattern of performance in different FoldFlow variants. More, specifically, we observe the following trends:
> > >
> > >  - Stochasticity improves robustness and the ability of the model to generate novel proteins in $7/8$ settings, as observed in the
> > >    fraction of novel and designable proteins (Novelty-fraction).
> > >  - Optimal transport improves the designability of the model by reducing the variance in the training objective in $7/8$ settings.
> > >  - The auxiliary losses improve the designability of the models in $7/8$ settings.
> > >  - Inference annealing improves the performance of all $8/8$ FoldFlow models in all metrics.
> > >
> > > The end of the discussion period is fast approaching and we would love to still have the opportunity to engage with you if there are any lingering questions. We thank you again for your time and valuable feedback, which have helped us make our experimental results more convincing.

---

> ### Comment · Area_Chair_HBCi · 2023-11-23
> **Please respond to the authors' rebuttal**
>
> Dear reviewer 4M3T,
>
> The author's have tried to address your concerns in their rebuttal. Note that the scores and reviews vary quite a bit for this submission. I'd like to understand how the rebuttal and the other reviews affect your review. Please respond to the rebuttal as soon as possible. The author-reviewer discussions are ending on Wednesday Nov 22 (in a few hours). The reviewer-AC discussions start afterwards until Dec 5th.
>
> Kind regards,
>
> Your AC

---

### Author Response · Authors · 2023-11-19
**Global Response to all Reviewers Part 1/3**

We thank all reviewers for their thorough reviews and valuable feedback. We are encouraged by the positive feedback our paper has received and appreciate the reviewers’ suggestions. We are happy to see that the reviewers found the work to be well-motivated (R-PWT4, R-Nm3A) and significant (R-zhj9) and the experimental validation to be thorough, promising and strong (R-zhj9, R-Nm3A, R-PWT4, R-PWT4). We are also heartened to hear that the reviewers found our work to be clear and well-written (R-zhj9, R-Nm3A). We now address the main shared concerns of the reviewers grouped by theme.

## Summary of new updates to the paper
To increase clarity and transparency all of our updates to the text of the uploaded submission PDF are colored in blue. Below we summarize the key changes.
1. **(R-zh9j, R-4M3T)** We have retrained FrameDiff with the same training set we used to train our models. These results are now included in Table 2, under FrameDiff-Retrained, and Fig. 9 (abc) in the appendix has also been updated to reflect this change. This model is now directly comparable to our FoldFlow models in terms of data, computational resources, and the majority of the hyperparameters used for training. We find that FoldFlow outperforms FrameDiff-Retrained on all metrics.

2. **(R-4M3T, R-PWT4, R-zh9j)** We have updated our main protein experiment results for FoldFlow-SFM using improved parameters for the stochastic inference/sampling procedure, which has resulted in increased performance across all metrics compared to the previously reported values. These updates are reflected in Table 2 as well as in Fig. 9 (abc) in the appendix.

3. **(R-4M3T, R-PWT4, R-zh9j)** We have included an additional metric to measure novelty, which measures the fraction of proteins that are designable (scRMSD < 2 A) and novel (max. TM-score to training set < 0.5), as used in [12]. Using this metric, we see that all FoldFlow models generate more novel and designable proteins compared to FrameDiff and that, in particular, FoldFlow-SFM outperforms both -OT and -BASE. This confirms our hypothesis about the generalization capabilities of the stochastic approach, which we further discuss below. We would also like to note that RFDiffusion and Genie are trained on a larger dataset, making their comparison based on “distance”  to our training dataset invalid. These results are included in Table 2 and Table 5 in the appendix.

4. **(R-zh9j, R-Nm3A)**  Table 2 has been updated to provide a complete picture of our results as it now includes both the average scRMSD values as well as the standard errors for the designability and novelty metrics. We note that due to the nature of the diversity metric (avg. pairwise TM-score for each sequence length in [100, 150, 200, 250, 300]), one cannot obtain a sensible uncertainty quantification without additional new full runs. We have also updated Appendix I to provide more details on our metrics and the associated error bounds.

5. **(R-zh9j, R-Nm3A)** We have added Table 5 in the appendix which explores additional ablations for model components, which confirm that inference annealing and auxiliary losses are also helpful for stochastic and OT models.

6. **(R-PWT4, R-zh9j)** We have included new baselines for the Equilibrium conformation generation experiment in Table 3 of the main paper. These baselines include FrameDiff and FoldFlow-Rand which both start from a random prior and lead to worse performance than FoldFlow which starts from the empirical data distribution.

Part 1/3

---

> ### Author Response · Authors · 2023-11-19
> **Global Response to all Reviewers Part 2/3**
>
> ## Motivations for SFM (R-4M3T, R-PWT4, R-zh9j)
> We first highlight that we have updated the main protein experiment results for FoldFlow-SFM, with improved parameters for stochastic inference. This update has resulted in improved performance of the model across all the metrics compared to the previously reported values including the original novelty metric as well as the new novelty metric that measures the fraction of generated samples that are $\textit{designable and novel}$. From the lens of novelty, this establishes a clear hierarchy of FoldFlow models with FoldFlow-SFM being the most novel model. Furthermore, these results add growing empirical validation to the claims in many past works that the added noise in SDEs—in comparison to ODEs— aids in learning more robust generative models in high dimensions [9,10,11]. For proteins, a proxy measure for this robustness is the ability to generate valid (designable) samples that are “out of distribution”, which corresponds to the training set. Both novelty metrics achieve this goal as they measure the fraction of designable proteins at various cutoff values for the max TM-score.
> From a biological point of view, we believe the design of novel proteins is a long-standing and ambitious goal for AI-powered drug design. It is well known that the design space of proteins is vast and the need for novel samples that are actually designable remains a challenging problem. Consequently, we argue benchmarking protein generative models in terms of their ability to generate designable yet novel samples in relation to the training set is an important goal. Towards this goal, FoldFlow-SFM produces the most novel samples compared to the previous SOTA non-pretrained diffusion model (FrameDiff-{ICML, Improved, Retrained}), as well as our FoldFlow-Base and FoldFlow-OT.
>
> ## Ablation Study (R-zh9j, R-NM3A)
> We thank the reviewers for suggesting these ablations, which have strengthened the empirical caliber of our paper. In Table 5 of the appendix, we have extended our ablation study of the features of the model (stochasticity, optimal transport, and inference annealing), to include more combinations as requested by the reviewers. We are also performing additional ablations with the Auxiliary loss, which require re-training the models and thus more time. We will report those results as soon as they become available as well.  In summary, we observe the following trends: inference annealing improves the performance of all FoldFlow, models in all metrics. Stochasticity improves robustness and the ability of the model to generate novel proteins. Optimal transport improves the designability of the model by reducing the variance in the training objective.
>
> ## Equilibrium Conformation Experiment (R-4M3T, R-PWT4, R-zh9j)
>
> We agree with the reviewers that our Equilibrium conformation generation experiments using FoldFlow can benefit from additional baselines that give the context needed to show the advantage of flow matching starting from an informed prior.
> To convincingly demonstrate this utility for equilibrium conformation generation, we now include $2$ baseline models: 1.) FrameDiff where the prior is a uniform distribution on $\text{SE}(3)^N_0$ 2.) and FoldFlow-Rand which is the FoldFlow-Base model where the prior is again a uniform distribution on $\text{SE}(3)^N_0$. For a fair comparison, both baselines were trained using the same training procedure and data as FoldFlow, our model that starts from the conformations obtained via folding models with added noise. We report the quantitative results of this extended experiment in the new Table 3 in the main paper. We find the following hierarchy of performance: FoldFlow (4.379) > FoldFlow-Rand (4.446) > FrameDiff (4.844), under the 2-Wasserstein distance over angles (lower is better). Additionally, we report the Wasserstein-2 distance for residue 56 for each model and find the same result—i.e. FoldFlow is the best followed by FoldFlow-Rand and FrameDiff.  These additional baseline results surface two important insights, namely FoldFlow convincingly outperforms diffusion, and using an informed prior—which is possible using a flow matching framework—leads to better results than an uninformed (random) prior holding everything else, including model class, constant. We thank the reviewers again for finding a deficiency in our previous experimental design and hope that the updated experimental findings help the reviewers endorse our paper more strongly.
>
>
> Part 2/3

---

> > ### Author Response · Authors · 2023-11-19
> > **Global Response to all Reviewers Part 3/3**
> >
> > ## References
> > [1] Kuhlman, B. & Bradley, P. Advances in protein structure prediction and design. Nat. Rev. Mol. Cell Biol. 20, 681–697 (2019).
> >
> > [2] Huang, P.-S., Boyken, S. E. & Baker, D. The coming of age of de novo protein design. Nature 537, 320–327 (2016).
> >
> > [3] Koga, N. et al. Principles for designing ideal protein structures. Nature 491, 222–227 (2012).
> >
> > [4] Cao, L. et al. Design of protein-binding proteins from the target structure alone. Nature 605, 551–560 (2022).
> >
> > [5] Kries, H., Blomberg, R. & Hilvert, D. De novo enzymes by computational design. Curr. Opin. Chem. Biol. 17, 221–228 (2013).
> >
> > [6] Joh, N. H. et al. De novo design of a transmembrane Zn2+-transporting four-helix bundle. Science 346, 1520–1524 (2014).
> >
> > [7] Smith, J. M. Natural selection and the concept of a protein space. Nature 225, 563–564 (1970).
> >
> > [8] Jumper, J. et al. Highly accurate protein structure prediction with AlphaFold. Nature 596, 583–589 (2021).
> >
> > [9] Alexander Tong, Nikolay Malkin, Kilian Fatras, Lazar Atanackovic, Yanlei Zhang, Guillaume Huguet, Guy Wolf, and Yoshua Bengio. Simulation-free schrödinger bridges via score and flow matching, 2023a.
> >
> > [10] Guan-Horng Liu, Arash Vahdat, De-An Huang, Evangelos A. Theodorou, Weili Nie, and Anima Anandkumar. I^2sb: Image-to-image schrödinger bridge.International Conference on Machine Learning (ICML), 2023a.
> >
> > [11] Yuyang Shi, Valentin De Bortoli, Andrew Campbell, and Arnaud Doucet. Diffusion Schrödinger bridge matching. arXiv preprint 2303.16852, 2023.
> >
> > [12] Yeqing Lin and Mohammed AlQuraishi. Generating novel, designable, and diverse protein structures by equivariantly diffusing oriented residue clouds, 2023.
> >
> > Part 3/3

---

### Comment · Area_Chair_HBCi · 2023-11-20
**To all reviewers: Please respond to the authors' rebuttal**

Dear reviewers,

The window for interacting with authors on their rebuttal is closing on Wednesday (Nov 21st). Please respond to the authors' rebuttal as soon as possible, so that you can discuss any agreements or disagreements. Note that the scores and reviews vary quite a bit for this submission, so it is crucial that you use this period to respond to the authors. Please acknowledge that you have read the authors' comments, and explain why their rebuttal does or does not change your opinion and score.

Many thanks,

Your AC

---

### Meta-Review · Area_Chair_HBCi · 2023-12-11

**Metareview:**

The reviewers unanimously support the acceptance of this paper. During the rebuttal phase the reviewers and authors engaged in constructive discussions, leading to additional results in the form of ablation studies and new baselines. These results and the discussions further strengthened the motivation of the proposed method. There was a high degree of consensus among the reviewers that the paper is well written, with interesting results, and that protein backbone generation is a timely application of flow-matching. Given the clear overall positive reception of this manuscript, the recommendation is to accept this paper with a spotlight presentation.

**Justification For Why Not Higher Score:**

In the current manuscript there is no generalization of the method to MD trajectories of proteins unseen during training. From my perspective, generalization is a crucial aspect in terms of the potential impact of this method, and therefore I would not consider it for an oral presentation without this.

**Justification For Why Not Lower Score:**

All reviewers are unanimously positive about the exciting contributions of this paper. Therefore I think the paper deserves a spotlight to distinguish it from other accepted papers that generated less excitement.

---

### Decision · Program_Chairs · 2024-01-16

Accept (spotlight)